# Diffusion Sampling Correction via Approximately 10 Parameters

**Guangyi Wang** [1 2 3 *]  **Wei Peng** [4 *]  **Lijiang Li** [1 3]  **Wenyu Chen** [5]  **Yuren Cai** [1 3]  **Songzhi Su** [1 3]

## Abstract

While powerful for generation, Diffusion Probabilistic Models (DPMs) face slow sampling challenges, for which various distillation-based methods have been proposed. However, they typically require significant additional training costs and model parameter storage, limiting their practicality. In this work, we propose **P**CA-based **A**daptive **S**earch (PAS), which optimizes existing solvers for DPMs with minimal additional costs. Specifically, we first employ PCA to obtain a few basis vectors to span the high-dimensional sampling space, which enables us to learn just a set of coordinates to correct the sampling direction; furthermore, based on the observation that the cumulative truncation error exhibits an "S"-shape, we design an adaptive search strategy that further enhances the sampling efficiency and reduces the number of stored parameters to approximately 10. Extensive experiments demonstrate that PAS can significantly enhance existing fast solvers in a plug-and-play manner with negligible costs. E.g., on CIFAR10, PAS optimizes DDIM's FID from 15.69 to 4.37 (NFE=10) using only **12 parameters and sub-minute training** on a single A100 GPU. Code is available at https://github.com/onefly123/PAS.

## 1. Introduction

Diffusion Probabilistic Models (DPMs) (Sohl-Dickstein et al., 2015; Ho et al., 2020; Song & Ermon, 2019; Song et al., 2021b; Karras et al., 2022) have demonstrated impressive generative capabilities in various fields, including image generation (Dhariwal & Nichol, 2021; Peebles

*Equal contribution [1]School of Informatics, Xiamen University, China [2]Xiamen Truesight Technology Co., Ltd, China [3]Key Laboratory of Multimedia Trusted Perception and Efficient Computing, Ministry of Education of China, Xiamen University, China [4]Stanford University, USA [5]Shandong University, China. Correspondence to: Songzhi Su <ssz@xmu.edu.cn>.

*Proceedings of the 42nd International Conference on Machine Learning*, Vancouver, Canada. PMLR 267, 2025. Copyright 2025 by the author(s).

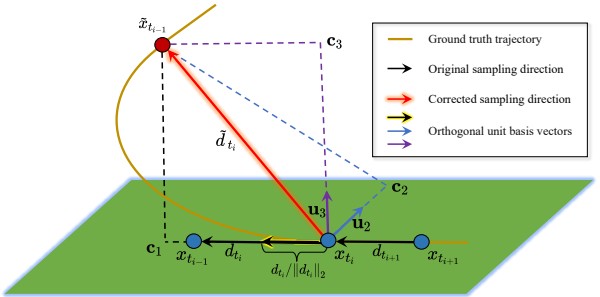

*Figure 1.* PCA-based sampling correction. We first utilize PCA to obtain a few orthogonal unit vectors that span the space of the sampling trajectories, and then learn the coordinates to correct the sampling directions in regions of large curvature along the ground truth trajectory.

& Xie, 2023), text-to-image generation (Rombach et al., 2022; Betker et al., 2023), video generation (Dehghani et al., 2023), and speech synthesis (Song et al., 2022), garnering widespread attention. DPMs introduce noise into the data through a forward process and then generate the actual output by iterative denoising during the reverse process. Compared to other generative models such as Generative Adversarial Networks (GANs) (Goodfellow et al., 2014) and Variational Autoencoders (VAEs) (Kingma & Welling, 2013), DPMs offer advantages in generating high-quality outputs and maintaining training stability. However, the denoising process in DPMs often requires hundreds or thousands of iterative steps, resulting in slow sampling speeds that severely hinder practical applications.

Existing sampling algorithms for accelerated DPMs can be categorized into two main categories: training-free methods and training-based methods. Training-free methods (Song et al., 2021a; Lu et al., 2022a;b; Bao et al., 2022b; Liu et al., 2022a; Karras et al., 2022; Zhao et al., 2023; Zheng et al., 2023; Zhang & Chen, 2023; Wang & Vastola, 2024; Wang et al., 2025) employ analytical solvers to reduce discretization errors per sampling iteration, achieving quality comparable to 1000-step sampling while requiring as few as 20 number of function evaluations (NFE). However, when NFE is less than 10, the accumulated truncation errors in these methods can be significantly magnified, leading to non-convergent sampling, which is ineffective and remains time-consuming. Training-based methods (Salimans & Ho,

2022; Liu et al., 2022b; Song et al., 2023; Luo et al., 2023; Yin et al., 2024) generally enhance sampling efficiency significantly, with the potential to achieve one-step sampling that rivals the quality of the original 1000 NFE. Nonetheless, these methods typically demand high computational resources and require additional model parameters storage. Even for the relatively simple CIFAR10 dataset, they may require over 100 A100 GPU hours (Salimans & Ho, 2022; Song et al., 2023), posing challenges for practical applications. Moreover, training-based methods often establish new paths between noise and data distributions, disrupting the interpolation capability between two disconnected modes.

To address these issues, we propose **P**CA-based **A**daptive **S**earch (PAS), a plug-and-play method that can correct the truncation errors of existing fast solvers with minimal training costs and a small number of learnable parameters. Additionally, PAS retains the interpolation capability between two disconnected modes. Inspired by previous observation that the sampling trajectories of DPMs lie in a low-dimensional subspace within a high-dimensional space (Zhou et al., 2024; Chen et al., 2024; Wang & Vastola, 2023; 2024), we propose employing Principal Component Analysis (PCA) to obtain a few orthogonal unit basis vectors in the low-dimensional subspace where the sampling trajectories lie, then learning the corresponding coefficients (i.e., coordinates) to determine the optimal sampling direction. This approach *avoids training neural networks to directly produce high-dimensional outputs*, significantly reducing the number of learnable parameters and training costs. Furthermore, we observe that the accumulated truncation errors of existing fast solvers exhibit an *"S"-shape*. We have designed an adaptive search strategy to balance the sampling steps that require correction and the truncation error. This further enhances the sampling efficiency of our method while reducing the amount of parameters required for storage. We validate the effectiveness of PAS on various unconditional and conditional pre-trained DPMs, across five datasets with resolutions ranging from 32 to 512. Results demonstrate that our method can significantly improve the image quality with negligible costs. Our contributions are summarized as follows:

- We propose a new plug-and-play training paradigm with about 10 parameters for existing fast DPMs solvers as an efficient alternative to the high-cost training-based algorithms, rendering the learnable parameters and training costs negligible.

- We design an adaptive search strategy to reduce correction steps, further enhancing the sampling efficiency of our method and decreasing the stored parameters.

- Extensive experiments across various datasets validate the effectiveness of the proposed PAS method in enhancing the sampling efficiency of existing fast solvers.

## 2. Background

### 2.1. Forward and Reverse Processes

The goal of diffusion probability models (DPMs) (Sohl-Dickstein et al., 2015; Ho et al., 2020; Song & Ermon, 2019; Song et al., 2021b; Karras et al., 2022) is to generate $D$-dimensional random variables $x_0 \in \mathbb{R}^D$ that follow the data distribution $q_{data}(x_0)$. DPMs add noise to the data distribution through a forward diffusion process; given $x_0$, the latent variables $\left\{ x_t \in \mathbb{R}^D \right\}_{t \in [0,T]}$ are defined as:

$$q(x_t \mid x_0) = \mathcal{N}(x_t \mid \alpha_t x_0, \sigma_t^2 \boldsymbol{I}), \tag{1}$$

where $\alpha_t \in \mathbb{R}$ and $\sigma_t \in \mathbb{R}$ are scalar functions related to the time step $t$. Furthermore, Song et al. (2021b) introduced stochastic differential equations (SDE) to model the process:

$$\mathrm{d}x_t = f(t)x_t \mathrm{d}t + g(t)\mathrm{d}w_t, \tag{2}$$

where $f(\cdot) : \mathbb{R} \to \mathbb{R}$, $g(\cdot) : \mathbb{R} \to \mathbb{R}$, and $w_t \in \mathbb{R}^D$ is the standard Wiener process (Oksendal, 2013). Put together Equation (1) and Equation (2), we can get $f(t) = \frac{\mathrm{d}\log\alpha_t}{\mathrm{d}t}$ and $g^2(t) = \frac{\mathrm{d}\sigma_t^2}{\mathrm{d}t} - 2\frac{\mathrm{d}\log\alpha_t}{\mathrm{d}t}\sigma_t^2$. Additionally, Song et al. (2021b) provided the corresponding reverse diffusion process from time step $T$ to 0 as follows:

$$\mathrm{d}x_t = \left[ f(t)x_t - g^2(t)\nabla_x \log q_t(x_t) \right] \mathrm{d}t + g(t)\mathrm{d}\bar{w}_t, \tag{3}$$

where $\nabla_x \log q_t(x_t)$ is referred to as the *score function*, which can be estimated through neural networks. Remarkably, Song et al. (2021b) proposed a *probability flow ordinary differential equation* (PF-ODE) with the same marginal distribution as Equation (3) at any time $t$ as follows:

$$\mathrm{d}x_t = \left[ f(t)x_t - \frac{1}{2}g^2(t)\nabla_x \log q_t(x_t) \right] \mathrm{d}t. \tag{4}$$

Unlike Equation (3), this PF-ODE does not introduce noise into the sampling process, making it a deterministic sampling procedure. Due to its simpler form and more efficient sampling, it is preferred in practical applications over SDE (Song et al., 2021b;a; Lu et al., 2022a).

### 2.2. Score Matching

To solve the PF-ODE in Equation (4), it is typically necessary to first employ a neural network $s_\theta(x_t, t)$ to estimate the unknown score function $\nabla_x \log q_t(x_t)$. The neural network $s_\theta$ is trained using the $L_2$ loss as follows:

$$\mathbb{E}_{x_0 \sim q_{data}} \mathbb{E}_{x_t \sim q(x_t|x_0)} \left\| s_\theta(x_t, t) - \nabla_x \log q_t(x_t) \right\|_2^2. \tag{5}$$

Additionally, Ho et al. (2020) proposed using a noise prediction network $\epsilon_\theta(x_t, t)$ to predict the noise added to $x_t$ at time step $t$. Other literature (Karras et al., 2022; Lu et al., 2022b) suggested using a data prediction network

$x_\theta(x_t, t)$ to directly predict $x_0$ at different time steps $t$. The relationship among these three prediction networks can be expressed as follows:

$$s_\theta(x_t, t) = -\frac{\epsilon_\theta(x_t, t)}{\sigma_t} = \frac{x_\theta(x_t, t) - x_t}{\sigma_t^2}. \qquad (6)$$

In this paper, we adopt the settings from EDM (Karras et al., 2022), specifically $f(t) = 0$, $g(t) = \sqrt{2t}$, derived from Equations (2) to (4), and $\alpha_t = 1$, $\sigma_t = t$ as stated in Equation (1). Furthermore, utilizing the noise prediction network $\epsilon_\theta$, Equation (4) can be expressed as:

$$\mathrm{d}x_t = \epsilon_\theta(x_t, t)\mathrm{d}t. \qquad (7)$$

According to the simple PF-ODE form in Equation (7), using the Euler-Maruyama (Euler) solver (Kloeden et al., 1992), the sampling process from $t_i$ to $t_{i-1}$ can be represented as:

$$x_{t_{i-1}} \approx x_{t_i} + (t_{i-1} - t_i)\epsilon_\theta(x_{t_i}, t_i), \qquad (8)$$

where $i \in [N, \cdots, 1]$ and $t_N = T, \cdots, t_0 = 0$.

## 3. The Proposed PAS Method

### 3.1. PCA-based Sampling Correction

Utilizing the Euler solver (Kloeden et al., 1992) with Equation (8) to approximate Equation (7) introduces notable discretization errors that can become significantly amplified with a limited number of iterations. The exact solution of Equation (7) is given by:

$$x_{t_{i-1}} = x_{t_i} + \int_{t_i}^{t_{i-1}} \epsilon_\theta(x_t, t)\mathrm{d}t. \qquad (9)$$

Let sampling direction $d_{t_i} := \epsilon_\theta(x_{t_i}, t_i)$. Existing fast solvers reduce discretization errors through various numerical approximations. For example, the PNDM (Liu et al., 2022a) employs linear multi-step methods, while the DPM-Solver (Lu et al., 2022a;b) utilizes Taylor expansion to correct the sampling direction $d_{t_i}$ in Equation (8) to approach the exact solution $\int_{t_i}^{t_{i-1}} \epsilon_\theta(x_t, t)\mathrm{d}t$. Training-based methods (Song et al., 2023; Salimans & Ho, 2022; Zhou et al., 2024) typically utilize neural networks to correct the direction $d_{t_i}$. In contrast to the aforementioned methods, we extract a few orthogonal unit basis vectors from the high-dimensional space of the sampling trajectory using PCA. By learning the coordinates corresponding to these basis vectors, we correct the direction $d_{t_i}$ to the optimal direction $\tilde{d}_{t_i}$, thereby minimizing the training cost.

Specifically, during the iteration process from $x_{t_i}$ to $x_{t_{i-1}}$, we first extract a set of basis vectors from the space of the existing sampling trajectories $\{x_{t_N}, \cdots, x_{t_i}\}$, where $t_N = T, \cdots, t_0 = 0$. A surprising finding is that when

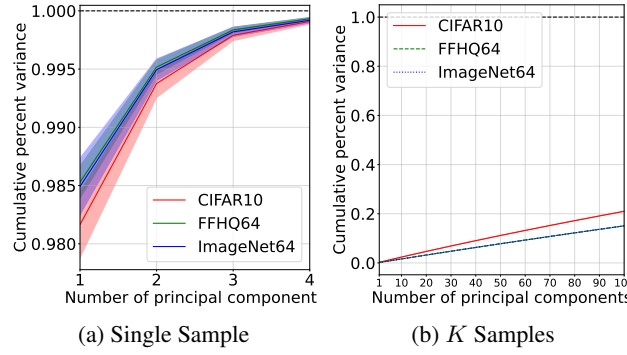

(a) Single Sample      (b) $K$ Samples

*Figure 2.* We utilize PCA to analyze the sampling trajectories, illustrating the trend of cumulative percent variance as the number of principal components varies. The trajectories are obtained from 1k samples using the Euler solver (Kloeden et al., 1992) in the EDM (Karras et al., 2022) pre-trained model with 100 NFE. (a) The average results of each trajectory $\{x_T, \{d_{t_i}\}_{i=N}^1\}$. (b) The results of $K$ trajectories $\{\{x_{t_i}^k\}_{i=N}^0\}_{k=1}^K$ (FFHQ and ImageNet curves nearly overlap).

performing PCA to decompose the entire sampling trajectory $\{x_{t_i}\}_{i=N}^0$, the cumulative percent variance saturates rapidly; by the time the number of principal components reaches 3, the cumulative percent variance approaches nearly 100%. This indicates that the entire sampling trajectory lies in a three-dimensional subspace embedded in a high-dimensional space, the finding initially revealed in works (Zhou et al., 2024; Chen et al., 2024; Wang & Vastola, 2023; 2024). Furthermore, according to Equation (8), $x_{t_{i-1}}$ is a linear combination of $x_{t_i}$ and $d_{t_i}$, allowing us to modify the existing trajectory $\{x_{t_N}, \cdots, x_{t_i}\}$ to $\{x_{t_N}, d_{t_N}, \cdots, d_{t_{i+1}}\}$. This modification enables our method to share buffers during the sampling process when combined with existing multi-step solvers that utilize historical gradients (*e.g.*, PNDM (Liu et al., 2022a), DEIS (Zhang & Chen, 2023), etc.) to optimize memory usage. To validate the reasonableness of this modification, we perform PCA on a complete sampling trajectory $\{x_T, \{d_{t_i}\}_{i=N}^1\}$, with the resulting cumulative percent variance illustrated in Figure 2a, showing that *three principal components suffice to span the space occupied by the entire sampling trajectory*. Notably, the sampling trajectories of different samples do not lie in the same three-dimensional subspace. We apply PCA to decompose the set of $K$ sampling trajectories from $K$ samples $\{\{x_{t_i}^k\}_{i=N}^0\}_{k=1}^K$, with the resulting cumulative percent variance displayed in Figure 2b. We observe that the cumulative percent variance does not show a saturation trend as the number of principal components increases. A theoretical analysis explaining why the sampling trajectories lie in low-dimensional subspaces, and why different samples occupy distinct subspaces, is provided in Section 3.4.

Based on this, during the iterative process from $x_{t_i}$ to $x_{t_{i-1}}$,

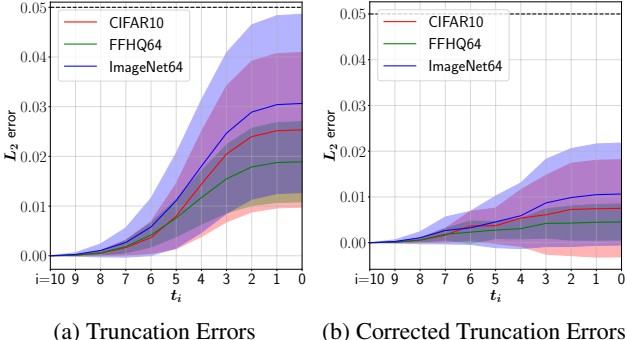

(a) Truncation Errors    (b) Corrected Truncation Errors

*Figure 3.* The truncation errors are evaluated using the Euler solver both with and without the proposed PAS. We utilize the EDM (Karras et al., 2022) pre-trained model to sample 10k samples and compute the average $L_2$ distance of 10 NFE compared to the ground truth trajectory (100 NFE). (a) The "S"-shaped truncation error is produced by the Euler solver. (b) The truncation error is corrected using PAS. Notably, PAS adaptively corrects only the parts of the sampling trajectory with large curvature.

we decompose the existing sampling trajectory, requiring only the top three basis vectors to span the space of the sampling trajectory. Let $\mathbf{X} = \{x_{t_N}, d_{t_N}, \cdots, d_{t_{i+1}}\}$, where $\mathbf{X} \in \mathbb{R}^{(N-i+1) \times D}$ and $D$ denotes the dimension of $x_{t_N}$. When using the top $k$ principal components, the process is described as:

$$\mathbf{W\Sigma V}^T = \text{SVD}(\mathbf{X}), \qquad (10)$$

$$\{\mathbf{v}_j\}_{j=1}^k = \mathbf{V}[:, :k], \qquad (11)$$

where SVD denotes the Singular Value Decomposition (SVD) and $\mathbf{v}_j \in \mathbb{R}^{D \times 1}$ represent orthogonal unit basis vectors. Further, since our goal is to correct the current direction $d_{t_i}$, we modify the above PCA process by directly specifying $\mathbf{v}_1 = d_{t_i}/\|d_{t_i}\|_2$. The subsequent approach generally involves computing the projection of $\mathbf{X}$ onto the basis vector $\mathbf{v}_1$, as follows:

$$\text{proj}_{\mathbf{v}_1}(\mathbf{X}) = \frac{\mathbf{X v}_1}{\|\mathbf{v}_1\|_2^2} \mathbf{v}_1^T. \qquad (12)$$

Then apply PCA to decompose $\mathbf{X} - \text{proj}_{\mathbf{v}_1}(\mathbf{X})$, obtaining the remaining two orthogonal unit basis vectors.

To further optimize computation time, we omit the projection step. After specifying $\mathbf{v}_1 = d_{t_i}/\|d_{t_i}\|_2$, we modify $\mathbf{X}$ as follows:

$$\mathbf{X}' = \text{Concat}(\mathbf{X}, d_{t_i}), \qquad (13)$$

where $\mathbf{X}' \in \mathbb{R}^{(N-i+2) \times D}$. Subsequently, we decompose $\mathbf{X}'$ using Equation (10) to obtain $\mathbf{V}'$, and then extract two new basis vectors, $\mathbf{v}_1', \mathbf{v}_2' = \mathbf{V}'[:, :2]$. Due to the omission of the projection step, the new basis vectors may be collinear with $\mathbf{v}_1$. Nevertheless, we only need to add one new basis vector $\mathbf{v}_3' = \mathbf{V}'[:, 2]$, sufficient to ensure that the sampling

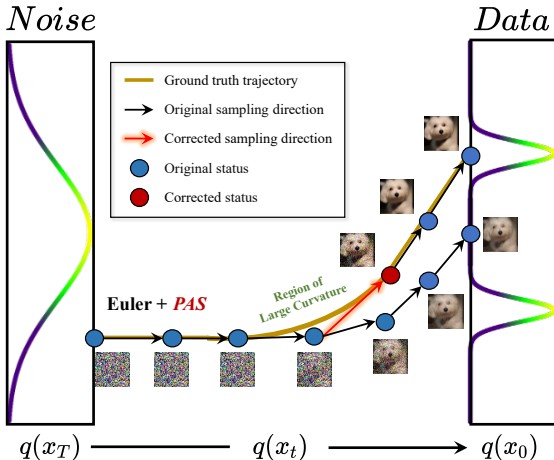

*Figure 4.* Illustration of PCA-based Adaptive Search (PAS). We demonstrate the Euler solver with the proposed PAS method, where sampling directions are derived from the tangent direction of the ground truth trajectory. The details of the correction process for the sampling direction are presented in Figure 1.

trajectory lies within the span of the basis vectors. Through Schmidt orthogonalization, we can obtain new orthogonal unit basis vectors as follows:

$$\mathbf{U} = [\mathbf{u}_1, \mathbf{u}_2, \mathbf{u}_3, \mathbf{u}_4] = \text{Schmidt}(\mathbf{v}_1, \mathbf{v}_1', \mathbf{v}_2', \mathbf{v}_3'), \quad (14)$$

where Schmidt represents the Schmidt orthogonalization and $\mathbf{U} \in \mathbb{R}^{D \times 4}$ consists of four orthogonal unit basis vectors. It is noteworthy that increasing a basis vector incurs less computational cost relative to the projection operation, and the additional single parameter can be considered negligible. After obtaining the basis vectors $\mathbf{U}$ that span the space of the sampling trajectory, we can initialize the learnable coordinate parameters. Since our goal is to correct the sampling direction $d_{t_i}$, and we have already specified the first basis vector $\mathbf{u}_1 = \mathbf{v}_1 = d_{t_i}/\|d_{t_i}\|_2$, we initialize the first coordinate as $\mathbf{c}_1 = \|d_{t_i}\|_2$, with the remaining coordinates initialized to zero, as follows:

$$\mathbf{C} = [\mathbf{c}_1 = \|d_{t_i}\|_2, \mathbf{c}_2 = 0, \mathbf{c}_3 = 0, \mathbf{c}_4 = 0]. \qquad (15)$$

At this point, we have $d_{t_i} = \mathbf{U C}^T$. Through training, we can obtain the optimized $\tilde{\mathbf{C}}$, thereby acquiring the corrected direction $\tilde{d}_{t_i} = \mathbf{U}\tilde{\mathbf{C}}^T$. The specific PCA-based sampling correction schematic is illustrated in Figure 1. In summary, we employ PCA to correct the sampling direction, requiring only a few sets of coordinates as learnable parameters. This approach serves as an efficient alternative to high-cost training-based algorithms, leveraging the geometric characteristics of the sampling trajectory in a low-dimensional space. As a result, it significantly reduces the number of learnable parameters and training costs.

---

**Algorithm 1** PCA-based Adaptive Search (PAS)

---

1: **Input:** initial value $x_T$, NFE $N$, model $\epsilon_\theta$, given solver $\phi$, time steps $\{t_i\}_{i=N}^0$, a ground truth trajectory $\{x_{t_i}^{gt}\}_{i=N}^0$, tolerance $\tau$
2: **def** PCA$(Q, d_{t_i})$:
3:     $\mathbf{W}\mathbf{\Sigma}\mathbf{V}^T = \mathrm{SVD}(\mathrm{Concat}(Q, d_{t_i}))$
4:     $\mathbf{v}_1 = d_{t_i}/\|d_{t_i}\|_2$; $\mathbf{v}_1', \mathbf{v}_2', \mathbf{v}_3' = \mathbf{V}[:, :3]$
5:     $\mathbf{u}_1, \mathbf{u}_2, \mathbf{u}_3, \mathbf{u}_4 = \mathrm{Schmidt}(\mathbf{v}_1, \mathbf{v}_1', \mathbf{v}_2', \mathbf{v}_3')$
6:     **Return:** $[\mathbf{u}_1, \mathbf{u}_2, \mathbf{u}_3, \mathbf{u}_4]$
7: $Q \xleftarrow{\mathrm{buffer}} x_T$, $d_{t_N} = \epsilon_\theta(x_T, t_N)$
8: **for** $i \leftarrow N$ **to** 1 **do**
9:   Init $\mathbf{c}_1 = \|d_{t_i}\|_2$, $\mathbf{c}_2 = 0$, $\mathbf{c}_3 = 0$, $\mathbf{c}_4 = 0$
10:   $\mathbf{C} = [\mathbf{c}_1, \mathbf{c}_2, \mathbf{c}_3, \mathbf{c}_4]$, $\mathbf{U} = \mathrm{PCA}(Q, d_{t_i})$
11:   $x_{t_{i-1}} = \phi(x_{t_i}, \mathbf{U}\mathbf{C}^T, t_i, t_{i-1})$
12:   $\tilde{\mathbf{C}} \leftarrow \mathbf{C} - \alpha\nabla_{\mathbf{C}}\|x_{t_{i-1}} - x_{t_{i-1}}^{gt}\|_2^2$
13:   $\tilde{x}_{t_{i-1}} = \phi(x_{t_i}, \mathbf{U}\tilde{\mathbf{C}}^T, t_i, t_{i-1})$
14:   $\mathcal{L}_1 = \|\tilde{x}_{t_{i-1}} - x_{t_{i-1}}^{gt}\|_2^2$, $\mathcal{L}_2 = \|x_{t_{i-1}} - x_{t_{i-1}}^{gt}\|_2^2$
15:   **if** $\mathcal{L}_2 - (\mathcal{L}_1 + \tau) > 0$ **then**
16:     coordinate_dict$[i] = \tilde{\mathbf{C}}$
17:     $x_{t_{i-1}} = \tilde{x}_{t_{i-1}}$, $d_{t_i} = \mathbf{U}\tilde{\mathbf{C}}^T$
18:   **end if**
19:   $Q \xleftarrow{\mathrm{buffer}} d_{t_i}$, $d_{t_{i-1}} = \epsilon_\theta(x_{t_{i-1}}, t_{i-1})$
20: **end for**
21: **Return:** coordinate_dict

---

**Algorithm 2** Sampling Correction

---

1: **Input:** initial value $x_T$, NFE $N$, model $\epsilon_\theta$, given solver $\phi$, time steps $\{t_i\}_{i=N}^0$, coordinate_dict
2: $Q \xleftarrow{\mathrm{buffer}} x_T$, $d_{t_N} = \epsilon_\theta(x_T, t_N)$
3: **for** $i \leftarrow N$ **to** 1 **do**
4:   **if** $i$ **in** coordinate_dict.keys() **then**
5:     $\mathbf{C} = $ coordinate_dict$[i]$, $\mathbf{U} = \mathrm{PCA}(Q, d_{t_i})$
6:     $d_{t_i} = \mathbf{U}\mathbf{C}^T$
7:   **end if**
8:   $x_{t_{i-1}} = \phi(x_{t_i}, d_{t_i}, t_i, t_{i-1})$
9:   $Q \xleftarrow{\mathrm{buffer}} d_{t_i}$, $d_{t_{i-1}} = \epsilon_\theta(x_{t_{i-1}}, t_{i-1})$
10: **end for**
11: **Return:** $x_{t_0}$

---

### 3.2. Training and Sampling

To correct the update direction $d_{t_i}$ during the iterative process from $x_{t_i}$ to $x_{t_{i-1}}$, we need to learn the coordinates $\mathbf{C}$ in Equation (15) to apply to the sampling trajectory of all samples. First, given any first-order ODE solver $\phi$, the discretized solution of Equation (9) can be uniformly represented as follows:

$$x_{t_{i-1}} = \phi(x_{t_i}, d_{t_i}, t_i, t_{i-1}), \tag{16}$$

where $d_{t_i} = \mathbf{U}\mathbf{C}^T = \epsilon_\theta(x_{t_i}, t_i)$. Given the ground truth $x_{t_{i-1}}^{gt}$, we can train the coordinates $\mathbf{C}$ using the stochastic gradient descent (SGD) algorithm (Robbins & Monro, 1951), with the $L_2$ loss update process as follows:

$$\tilde{\mathbf{C}} \leftarrow \mathbf{C} - \alpha\nabla_{\mathbf{C}}\|x_{t_{i-1}} - x_{t_{i-1}}^{gt}\|_2^2, \tag{17}$$

where $\alpha$ denotes the learning rate, and the specific acquisition method for $x_{t_{i-1}}^{gt}$ is discussed in Section 3.3. After training $\mathbf{C}$ using multiple samples through Equation (17), we obtain the trained coordinates $\tilde{\mathbf{C}}$.

During the iterative process from $x_{t_i}$ to $x_{t_{i-1}}$, by utilizing the trained coordinates $\tilde{\mathbf{C}}$, we can correct the current update direction $d_{t_i}$ to $\tilde{d}_{t_i} = \mathbf{U}\tilde{\mathbf{C}}^T$, thereby obtaining a more accurate $\tilde{x}_{t_{i-1}}$, as follows:

$$\tilde{x}_{t_{i-1}} = \phi(x_{t_i}, \tilde{d}_{t_i}, t_i, t_{i-1}). \tag{18}$$

### 3.3. Adaptive Search

In Sections 3.1 and 3.2, we introduced how to correct the iterative process from $x_{t_i}$ to $x_{t_{i-1}}$ using our method. This section describes how to correct the iterative process from $x_T$ to $x_0$ using our approach. First, we need to generate a ground truth trajectory $\{x_{t_i}^{gt}\}_{i=N}^0$ to correct $\{x_{t_i}\}_{i=N}^0$, where $x_{t_N}^{gt} = x_{t_N}$. In this paper, we adopt a widely used polynomial time schedule (Karras et al., 2022) for both sampling and generating the ground truth trajectory, which is expressed as follows:

$$t_i = (t_0^{1/\rho} + \frac{i}{N}(t_N^{1/\rho} - t_0^{1/\rho}))^\rho, \; i \in [N, \cdots, 0], \quad (19)$$

where $t_N = T, \cdots, t_0 = \epsilon$, and $\epsilon$ is a value approaching zero. To obtain the ground truth trajectory, we simply need to insert more sampling steps into the time schedule from Equation (19) to achieve a more accurate solution. Specifically, consider using a teacher Euler solver with $N'$ NFE to guide a student Euler solver with $N(< N')$ NFE during training. First, we insert $M$ values into the time schedule for the student solver, such that $M$ is the smallest positive integer satisfying $N(M + 1) \geq N'$. Next, We use Equation (19) to generate the time schedule for the teacher solver: $t_{N(M+1)} = T, \cdots, t_0 = \epsilon$. Finally, we only need to index the $x_{t_{i(M+1)}}$ from the teacher solver using the $i \in [N, \cdots, 0]$, thereby obtaining the ground truth trajectory $\{x_{t_i}^{gt}\}_{i=N}^0 = \{x_{t_{i(M+1)}}\}_{i=N}^0$.

After obtaining the ground truth trajectory $\{x_{t_i}^{gt}\}_{i=N}^0$, we need to sequentially correct $d_{t_N}, \cdots, d_{t_1}$. This is because once $d_{t_N}$ is corrected to $\tilde{d}_{t_N}$, $x_{t_{N-1}}$ will be adjusted accordingly to $\tilde{x}_{t_{N-1}}$. This further modifies the next time point direction that requires correction: $d_{t_{N-1}} = \epsilon_\theta(\tilde{x}_{t_{N-1}}, t_{N-1})$. In general, we need to correct $N$ directions sequentially, storing $4N$ learned coordinate parameters, and correcting $N$ iterative processes during sampling. Nevertheless, to further reduce the additional computational cost during the sampling process and the number of stored coordinate pa-

rameters, we propose an adaptive search strategy. Specifically, the cumulative truncation error of the existing solvers exhibits an *"S"-shaped trend*, as shown in Figure 3a, indicating that it initially grows slowly, then increases rapidly, and ultimately returns to a slow growth rate. Thus, we can infer that the sampling trajectory first appears linear, then transitions to a curve, and ultimately becomes linear again under the attraction of a certain mode. Consequently, *only the parts of the sampling trajectory with large curvature require correction*; the linear sections do not. We employ PCA to obtain the basis of the space containing the sampling trajectory, also aiming to compensate for the missing directions in other bases due to discretization of Equation (9) in cases of large curvature. The specific implementation of the adaptive search is determined by the loss of the optimized state. When using $L_2$ loss, we obtain:

$$\mathcal{L}_1 = \|\tilde{x}_{t_{i-1}} - x_{t_{i-1}}^{gt}\|_2^2, \ \mathcal{L}_2 = \|x_{t_{i-1}} - x_{t_{i-1}}^{gt}\|_2^2, \quad (20)$$

where $\tilde{x}_{t_{i-1}}$ is the corrected state. We introduce a tolerance $\tau$ to determine whether $\mathcal{L}_2 - (\mathcal{L}_1 + \tau)$ is greater than zero. If it is greater than zero, correction is required for that step; otherwise, the step is considered to lie within the linear part of the sampling trajectory, and no correction is necessary. The tolerance $\tau$ is set to a positive value, *e.g.* $10^{-4}$. The truncation error after correction using our algorithm is depicted in Figure 3b, clearly showing a significant reduction in truncation error in the large curvature regions. Now that we have thoroughly presented the proposed PCA-based Adaptive Search (PAS) algorithm, detailing the complete training and sampling processes in Algorithms 1 and 2. The specific schematic is illustrated in Figure 4.

### 3.4. Theoretical Insights and Discussion

We attribute the efficiency of the proposed PAS to several key observations about the geometric structure of sampling trajectories: First, the sampling trajectories of the samples lie in a low-dimensional subspace (Figures 2a and 6c), which enables PAS to correct the sampling direction effectively by learning only a few sets of low-dimensional coordinates corresponding to the basis vectors of the sampling subspace. Second, although the sampling trajectories of different samples lie in distinct subspaces (Figure 2b), their geometric shapes exhibit strong consistency (Figures 3a and 6d). The correction coordinates learned by PAS at each time step $t_i$ implicitly capture the curvature information of sample trajectories. In other words, for a given dataset, PAS leverages the strongly consistent geometric characteristics learned from a small number of samples and generalizes them to all samples within the dataset, thereby achieving efficient sampling correction and acceleration.

We have already empirically validated the aforementioned key observations through experiments and prior studies in Sections 3 and 4 and Appendix A. Next, we provide a heuris-

tic theoretical analysis to explain why these key observations hold. Inspired by Wang et al. (2023; 2024), and acknowledging that directly analyzing neural networks is inherently intractable, Wang et al. (2023; 2024) derived an analytical form of Gaussian score structures to approximate diffusion trajectories, and provided both theoretical and empirical support. For the EDM (Karras et al., 2022) trajectories, an approximate analytical form is given:

$$x_t = \mu + \frac{\sigma_t}{\sigma_T} x_T^{\perp} + \sum_{k=1}^{r} \psi(t, \lambda_k) c_k(T) u_k, \quad (21)$$

where $\mu$ and $u_k$ represent the dataset-dependent mean and basis vectors that define the data manifold, and $x_T^{\perp}$ denotes the off-manifold component. $\sigma_t$ and $\psi(t, \lambda_k)$ are functions dependent on time $t$. Given $x_T$, both $\sigma_T$ and $c_k(T)$ are constants. From Equation (21), it is evident that $x_t$ is a linear combination of the vectors $x_T^{\perp}$, $\mu$, and $u_k$. For a fixed dataset, the rate of change of the coefficients for the vectors $x_T^{\perp}$, $\mu$, and $u_k$ depends only on the timestep $t$. As time progresses, all samples evolve along their respective off-manifold directions toward the data manifold with the same scaling. This theory heuristically explains why all samples exhibit strongly consistent geometric characteristics. Furthermore, Wang et al. (2023) theoretically derived that the diffusion trajectories resemble 2D rotations on the plane spanned by the initial noise $x_T$ and the final sample $x_0$. Since $x_T$ varies across different samples, the corresponding planes also differ, providing a theoretical underpinning for why the trajectories of different samples lie in distinct subspaces. At the same time, this also theoretically supports the observation that the sampling trajectories of the samples lie in a low-dimensional (2D) subspace.

### 3.5. Comparing with Training-based Methods

As discussed in Section 1, while training-based methods can achieve one-step sampling (Salimans & Ho, 2022; Liu et al., 2022b; Song et al., 2023; Yin et al., 2024), they often incur substantial training costs (*e.g.*, exceeding 100 A100 GPU hours on simple CIFAR10). Moreover, distillation-based methods tend to disrupt the original ODE trajectories, resulting in the loss of interpolation capability between two disconnected modes, thereby limiting their effectiveness in downstream tasks that rely on such interpolation. Although some low-cost training methods (Kim et al., 2023a; Hsiao et al., 2024; Bao et al., 2022a; Kim et al., 2023b; Na et al., 2024; Zhang et al., 2024; Zhou et al., 2024) have been proposed, as discussed in Appendix A, these methods still do not address the essential issue and require training a new, relatively small neural network.

In contrast to these methods, PAS introduces a new training paradigm that *corrects high-dimensional vectors by learning low-dimensional coordinates*, achieving minimal learnable

*Table 1.* On the CIFAR10, the list of numbers denotes all the time points requiring correction from adaptive search for the DDIM and iPNDM solvers, with each element $i$ ranging from NFE ($N$) to 1.

| Method | NFE | | | |
|---|---|---|---|---|
| | 5 | 6 | 8 | 10 |
| DDIM + PAS | 3,1 | 4,2,1 | 5,3,2 | 6,4,2 |
| iPNDM + PAS | 2 | 3 | 3,1 | 4,2 |

parameters and training costs. For instance, using a single NVIDIA A100 GPU, training on CIFAR10 takes only **0~2 minutes**, and merely **10~20 minutes** on datasets with a maximum resolution of 256. Notably, the computational cost for PCA is negligible compared to that of NFE. For example, in Stable Diffusion v1.4, one NFE takes approximately 30.2 seconds for 128 samples, whereas one PCA computation takes only 0.06 seconds. Additionally, based on adaptive search, PAS only requires correcting 1~3 time points on the CIFAR10, as shown in Table 1 (results for additional datasets, see Table 6 in Appendix C.1). This means that PAS only requires **4~12 parameters** during the sampling correction process. This is not in the same order of magnitude as the aforementioned training-based methods. Furthermore, PAS preserves the original ODE trajectories, thereby retaining the interpolation capability of DPMs.

## 4. Experiments

To validate the effectiveness of PAS as a plug-and-play and low-cost training method, we conducted extensive experiments on both conditional (Karras et al., 2022; Rombach et al., 2022) and unconditional (Karras et al., 2022; Song et al., 2023) pre-trained models.

### 4.1. Settings

In this paper, we adopt the design from the EDM framework (Karras et al., 2022) uniformly, as shown in Equation (7). Regarding the time schedule, to ensure a fair comparison, **we consistently utilize the widely used polynomial schedule with $\rho = 7$ in Table 2**, as described in Equation (19).

**Datasets and pre-trained models.** We employ PAS across a wide range of images with various resolutions (from 32 to 512). These include CIFAR10 32×32 (Krizhevsky et al., 2009), FFHQ 64×64 (Karras et al., 2019), ImageNet 64×64 (Deng et al., 2009), LSUN Bedroom 256×256 (Yu et al., 2015), and images generated by Stable Diffusion v1.4 (Rombach et al., 2022) with 512 resolution. Among these, the CIFAR10, FFHQ, and LSUN Bedroom datasets are derived from the pixel-space unconditional pre-trained models (Karras et al., 2022; Song et al., 2023); the ImageNet comes from the pixel-space conditional pre-trained model (Karras et al., 2022); and the Stable Diffusion

v1.4 (Rombach et al., 2022) belongs to the conditional latent-space pre-trained model.

**Solvers.** We provide comparative results from previously state-of-the-art fast solvers, including DDIM (Song et al., 2021a), DPM-Solver-2 (Lu et al., 2022a), DPM-Solver++ (Lu et al., 2022b), DEIS-tAB3 (Zhang & Chen, 2023), UniPC (Zhao et al., 2023), DPM-Solver-v3 (Zheng et al., 2023), and improved PNDM (iPNDM) (Liu et al., 2022a; Zhang & Chen, 2023).

**Evaluation.** We evaluate the sample quality using the widely adopted Fréchet Inception Distance (FID↓) (Heusel et al., 2017) metric. For Stable Diffusion, we sample 10k samples from the MS-COCO (Lin et al., 2014) validation set to compute the FID, while for other datasets, we uniformly sample 50k samples.

**Training.** In Section 4.3 and Appendix C.2, we present ablation experiments related to the hyperparameters involved in training, and Appendix B provides detailed training configurations for different datasets that we used. Below, we outline some recommended settings for training hyperparameters: utilizing Heun's 2nd solver (Karras et al., 2022) from EDM to generate 5k ground truth trajectories with 100 NFE, employing the $L_1$ loss function, setting the learning rate to $10^{-2}$, and using a tolerance $\tau$ of $10^{-4}$.

### 4.2. Main Results

In this section, we present the experimental results of PAS across various datasets and pre-trained models with NFE $\in \{5, 6, 8, 10\}$. In Table 2, we report the experimental results of PAS correcting DDIM and iPNDM solvers, covering the CIFAR10, FFHQ, ImageNet, and LSUN Bedroom datasets. Notably, for the LSUN Bedroom, the order of iPNDM is set to 2; for the other datasets, the order is set to 3, which yields better average performance (for more results regarding the order of iPNDM, see Appendix C.3). Experimental results demonstrate that PAS consistently improves the sampling quality of both DDIM and iPNDM solvers, regardless of image resolution (large or small) or model type (conditional or unconditional pre-trained). Particularly, PAS combined with iPNDM surpasses the previously state-of-the-art solvers. Furthermore, we compared PAS with previous works that enhance sampling efficiency through trajectory regularity on the CIFAR10 and FFHQ datasets, including AMED (Zhou et al., 2024) and GITS (Chen et al., 2024). The results demonstrate that PAS provides a more significant improvement. To further explore the potential of PAS, we employ the analytical approach proposed by Wang et al. (2023) to "warm up" the basis vectors from the Gaussian score structure, and then perform sampling correction starting from the teleported solution. The results combining *teleportation* (TP) with $\sigma_{skip} = 10.0$ and PAS are presented in Table 2, demonstrating the enhanced performance and

*Table 2.* Sample quality measured by Fréchet Inception Distance (FID↓) on CIFAR10, FFHQ, ImageNet, and LSUN Bedroom datasets. "\" indicates missing data due to the inherent characteristics of the algorithm.

| Method | NFE | | | |
|---|---|---|---|---|
| | 5 | 6 | 8 | 10 |
| **CIFAR10 32×32** (Krizhevsky et al., 2009) | | | | |
| DDIM (Song et al., 2021a) | 49.68 | 35.63 | 22.32 | 15.69 |
| DDIM + AMED (Zhou et al., 2024) | \ | 25.15 | 17.03 | 11.33 |
| DDIM + GITS (Chen et al., 2024) | 28.05 | 21.04 | 13.30 | 10.37 |
| DDIM + TP (Wang & Vastola, 2024) | 24.50 | 18.41 | 12.04 | 8.78 |
| DDIM + PAS (**Ours**) | 17.13 | 12.11 | 7.07 | 4.37 |
| DDIM + TP + PAS (**Ours**) | **9.15** | **5.16** | **3.65** | **3.16** |
| DPM-Solver-2 (Lu et al., 2022a) | \ | 60.00 | 10.30 | 5.01 |
| DPM-Solver++(3M) (Lu et al., 2022b) | 31.65 | 17.89 | 8.30 | 5.16 |
| DEIS-tAB3 (Zhang & Chen, 2023) | 17.65 | 11.84 | 6.82 | 5.64 |
| UniPC(3M) (Zhao et al., 2023) | 31.44 | 17.74 | 8.42 | 5.31 |
| iPNDM (Zhang & Chen, 2023) | 16.55 | 9.74 | 5.23 | 3.69 |
| iPNDM + TP (Wang & Vastola, 2024) | 7.25 | 4.89 | 3.08 | 2.49 |
| iPNDM + PAS (**Ours**) | 13.61 | 7.47 | 3.87 | 2.84 |
| iPNDM + TP + PAS (**Ours**) | **5.16** | **3.76** | **2.77** | **2.40** |
| **FFHQ 64×64** (Karras et al., 2019) | | | | |
| DDIM (Song et al., 2021a) | 43.92 | 35.21 | 24.38 | 18.37 |
| DDIM + AMED (Zhou et al., 2024) | \ | 32.46 | 20.72 | 15.52 |
| DDIM + GITS (Chen et al., 2024) | 29.80 | 23.67 | 16.60 | 13.06 |
| DDIM + TP (Wang & Vastola, 2024) | 26.61 | 21.34 | 15.11 | 11.69 |
| DDIM + PAS (**Ours**) | 29.07 | 17.63 | 8.16 | 5.61 |
| DDIM + TP + PAS (**Ours**) | **10.37** | **7.37** | **4.54** | **4.50** |
| DPM-Solver-2 (Lu et al., 2022a) | \ | 83.17 | 22.84 | 9.46 |
| DPM-Solver++(3M) (Lu et al., 2022b) | 23.50 | 14.93 | 9.58 | 6.96 |
| DEIS-tAB3 (Zhang & Chen, 2023) | 19.47 | 11.61 | 8.64 | 7.07 |
| UniPC(3M) (Zhao et al., 2023) | 22.82 | 14.30 | 10.07 | 7.39 |
| iPNDM (Zhang & Chen, 2023) | 17.26 | 11.31 | 6.82 | 4.95 |
| iPNDM + TP (Wang & Vastola, 2024) | 10.26 | 7.28 | 4.83 | 3.94 |
| iPNDM + PAS (**Ours**) | 15.89 | 10.29 | 5.85 | 4.28 |
| iPNDM + TP + PAS (**Ours**) | **8.20** | **5.49** | **3.99** | **3.44** |
| **ImageNet 64×64** (Deng et al., 2009) | | | | |
| DDIM (Song et al., 2021a) | 43.81 | 34.03 | 22.59 | 16.72 |
| DDIM + PAS (**Ours**) | 31.37 | 26.21 | 12.33 | 9.13 |
| DPM-Solver-2 (Lu et al., 2022a) | \ | 44.83 | 12.42 | 6.84 |
| DPM-Solver++(3M) (Lu et al., 2022b) | 27.72 | 17.18 | 8.88 | 6.44 |
| DEIS-tAB3 (Zhang & Chen, 2023) | 21.06 | 14.16 | 8.43 | 6.36 |
| UniPC(3M) (Zhao et al., 2023) | 27.14 | 17.08 | 9.19 | 6.89 |
| iPNDM (Zhang & Chen, 2023) | **19.75** | 13.48 | 7.75 | 5.64 |
| iPNDM + PAS (**Ours**) | 23.33 | **12.89** | **7.27** | **5.32** |
| **LSUN Bedroom 256×256** (Yu et al., 2015) (pixel-space) | | | | |
| DDIM (Song et al., 2021a) | 34.34 | 25.25 | 15.71 | 11.42 |
| DDIM + PAS (**Ours**) | 28.22 | 13.31 | 7.26 | 6.23 |
| DPM-Solver-2 (Lu et al., 2022a) | \ | 80.59 | 23.26 | 9.61 |
| DPM-Solver++(3M) (Lu et al., 2022b) | 17.39 | 11.97 | 9.86 | 7.11 |
| DEIS-tAB3 (Zhang & Chen, 2023) | 16.31 | 11.75 | 7.00 | 5.18 |
| UniPC(3M) (Zhao et al., 2023) | 17.29 | 12.73 | 11.91 | 7.79 |
| iPNDM (Zhang & Chen, 2023) | 18.15 | 12.90 | 7.98 | 6.17 |
| iPNDM + PAS (**Ours**) | **15.48** | **10.24** | **6.67** | **5.14** |

*Table 3.* Sample quality measured by FID↓ on Stable Diffusion v1.4 with a guidance scale of 7.5. [†]We borrow the results reported in Zheng et al. (2023) directly.

| Method | NFE | | | |
|---|---|---|---|---|
| | 5 | 6 | 8 | 10 |
| DDIM (Song et al., 2021a) | 23.42 | 20.08 | 17.72 | 16.56 |
| [†]DPM-Solver++ (Lu et al., 2022b) | 18.87 | 17.44 | 16.40 | 15.93 |
| [†]UniPC (Zhao et al., 2023) | 18.77 | 17.32 | 16.20 | 16.15 |
| [†]DPM-Solver-v3 (Zheng et al., 2023) | 18.83 | 16.41 | 15.41 | 15.32 |
| DDIM + PAS (**Ours**) | **17.70** | **15.93** | **14.74** | **14.23** |

PAS when combined with DDIM surpasses the sampling results of previous state-of-the-art methods, further validating the effectiveness of the proposed PAS.

In Figure 5 and Appendix C.4, we present the visualization results for Stable Diffusion, as well as for the CIFAR10, FFHQ, ImageNet, and Bedroom datasets. The results indicate that the samples generated by PAS exhibit *higher quality and richer detail*. Through the experiments above, we have demonstrated that PAS serves as a plug-and-play, low-cost training method that can effectively enhance the performance of existing fast solvers (including DDIM and iPNDM), thereby validating the effectiveness of PAS.

### 4.3. Ablation Study

In this section, we conduct ablation experiments on several key modules used during the training process, as illustrated in Figure 6. Additionally, ablation experiments on the learning rate, the solver for generating trajectories, and the tolerance $\tau$ are presented in Appendix C.2. Notably, the selection of these modules in the overall PAS is not a critical factor and only has a slight impact on performance.

**Adaptive search.** In Figure 6a, we present the experimental results comparing the PAS with and without the adaptive search strategy (-AS). The findings reveal that the sampling quality of the PAS(-AS) is even inferior to DDIM. This degradation may be attributed to the linear segments in the sampling trajectory, where the errors from DDIM are negligible. However, the PCA-based sampling correction does not reduce error and instead introduces biases in other basis vectors. This further validates the necessity and effectiveness of the proposed *overall* PAS.

**Loss function.** We evaluated the $L_1$, $L_2$, and previously established effective loss functions: LPIPS (Zhang et al., 2018) and Pseudo-Huber (Song & Dhariwal, 2024). Here, the hyperparameter $c$ for the Pseudo-Huber was set to 0.03, as recommended by Song et al. (2024). The results are presented in Figure 6b. Surprisingly, LPIPS exhibits the lowest average performance, possibly because it was trained on clean images rather than noisy ones. Overall, the $L_1$ loss function demonstrates superior average performance, which

strong plug-and-play potential of PAS. For Stable Diffusion, we report the experimental results of PAS correcting DDIM. Additionally, we introduce previous state-of-the-art methods, including DPM-Solver++, UniPC, and DPM-Solver-v3, for comparison. These methods utilize their optimal configurations on Stable Diffusion, including the 2M version and logSNR schedule (Lu et al., 2022a), etc. As shown in Table 3, PAS significantly improves the sampling quality of DDIM on Stable Diffusion. Notably, the performance of

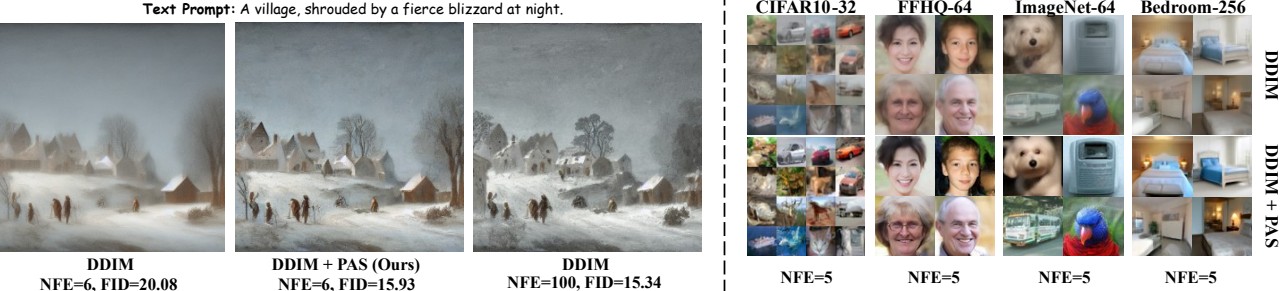

*Figure 5.* Visualization results using DDIM with and without the proposed PAS. Left: Sampling results on Stable Diffusion v1.4 with a guidance scale of 7.5. Right: Sampling results on the CIFAR10, FFHQ 64×64, ImageNet 64×64, and LSUN Bedroom 256×256 datasets.

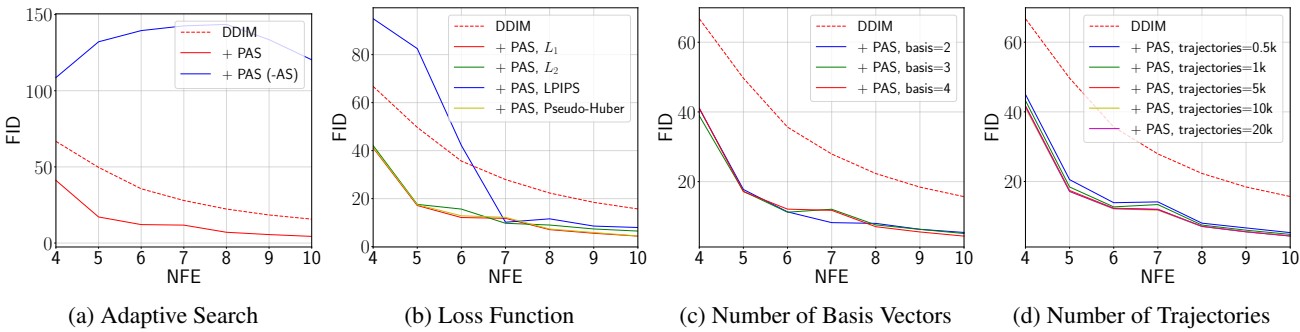

    (a) Adaptive Search        (b) Loss Function        (c) Number of Basis Vectors        (d) Number of Trajectories

*Figure 6.* Ablation study on CIFAR10, utilizing PAS to correct DDIM, exploring the impact of adaptive search, loss function, the number of orthogonal unit basis vectors, and the number of ground truth trajectories on FID (recommended setting: the red solid line).

may be attributed to its larger scale.

**Number of basis vectors.** We demonstrate the ablation results in Figure 6c, by varying the number of basis vectors used. Experimental results indicate that PAS can significantly improve the sampling quality of DDIM using only the top 2 basis vectors, while employing the top 3 or 4 vectors yields slightly better performance. Notably, the experimental results in Figure 6c exhibit the same trend as those in Figure 2a, further validating that the sampling trajectory of DPMs lies in a low-dimensional subspace. Four basis vectors suffice to span the sampling trajectory space, enabling PAS to achieve minimal training costs and learnable parameters compared to other training-based algorithms.

**Number of trajectories.** In Figure 6d, we vary the number of ground truth trajectories from 500 to 20k. We find that learning coordinates from as few as 500 trajectories can significantly enhance the sampling quality of DDIM. This demonstrates that the sampling trajectories of all samples exhibit *strong consistent geometric characteristics*, specifically the "S"-shaped truncation error. This also explains why a set of coordinates can effectively adapt to all samples within a single dataset. However, increasing the number of trajectories generally results in more generalized learned coordinates, with 5k trajectories being the optimal balance.

## 5. Conclusion

In this paper, we introduced a novel training paradigm, *PAS*, for accelerating DPMs with minimal training costs and learnable parameters. Our key strategy is to obtain a few basis vectors via PCA, and then learn their low-dimensional coordinates to correct the high-dimensional sampling direction vectors. Moreover, based on the observation that the truncation error of existing fast solvers exhibits an "S"-shape, we design an adaptive search strategy to balance the correction steps, further enhancing sampling efficiency and reducing the number of stored parameters to approximately 10. Extensive experiments on both unconditional and conditional pre-trained DPMs demonstrate that PAS can significantly improve the sampling quality of existing fast solvers, such as DDIM and iPNDM, in a plug-and-play manner.

In future work, we plan to further investigate the regularity of diffusion trajectories, focusing on leveraging their geometric characteristics to enable rapid or even one-step sampling at minimal cost. For instance, given the low dimensionality and strong consistency in the geometric shapes of sampling trajectories across different samples, it is promising to learn the integrated form of analytical trajectory equations from a small number of samples, thereby enabling one-step sampling with minimal cost.

## Impact Statement

Diffusion models can generate images, audio, and video content that is indistinguishable from human-created media. This work proposes an accelerated sampling algorithm based on the geometric characteristics of the diffusion model's sampling trajectory. However, this method could potentially be misused, accelerating the generation of malicious content, such as fake news or manipulated images, leading to detrimental societal impacts. We are fully aware of this potential risk and plan to focus on the discrimination of generated content in future research to mitigate its possible negative effects on society.

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

## A. Related Works

**Trajectory-based methods.** Recently, several works (Zhou et al., 2024; Chen et al., 2024; Wang & Vastola, 2023; 2024) have designed efficient sampling methods by analyzing the geometric properties of the sampling trajectories of diffusion models. For instance, Zhou et al. (2024) observed that the sampling trajectories of diffusion models lie in low-dimensional subspaces, thereby utilizing the Mean Value Theorem to reduce the output dimensions of neural networks and optimize training costs. Chen et al. (2024) found that the shape of the sampling trajectories exhibits strong consistency, and based on the shape of the trajectories, they employed dynamic programming algorithms to design more efficient sampling schedules. The proposed PAS method builds upon these in-depth studies of the geometric properties of sampling trajectories, particularly the characteristic that the trajectories of diffusion models lie within a low-dimensional subspace embedded in high-dimensional space. In contrast to the aforementioned methods, PAS innovatively proposes using PCA to obtain a few basis vectors of the sampling space, and further *corrects high-dimensional vectors by learning their low-dimensional coordinates*, thereby significantly reducing additional computational costs. Furthermore, we observe that the truncation errors of existing fast solvers exhibit an "S"-shape, leading to the proposal of an adaptive search strategy to further reduce overhead.

**Low-cost training.** Previous studies (Salimans & Ho, 2022; Liu et al., 2022b; Song et al., 2023) have shown that directly learning the mapping between noise and data distributions necessitates high training costs for minimal-step sampling. Recently, several low-cost training methods have been proposed. Kim et al. (2023a) and Hsiao et al. (2024) explored how to reduce the number of parameters in student models to achieve efficient distillation. Bao et al. (2022a), Kim et al. (2023b), and Na et al. (2024) corrected errors arising during the sampling process by training smaller neural networks. Zhang et al. (2024) suggested training the neural network only for the last step of sampling to eliminate accumulated residuals, thereby reducing training costs. However, these methods typically still require training a new, relatively small neural network. Unlike these approaches, the proposed PAS method requires learning only a few sets of coordinates, which results in *minimized learnable parameters and training costs*.

**Plug-and-play acceleration.** Numerous studies (Ma et al., 2024; Wimbauer et al., 2024; Li et al., 2023; Chen et al., 2024; Shih et al., 2023; Si et al., 2024; Xia et al., 2024; Wang et al., 2025) have explored ways to accelerate existing fast solvers, such as DDIM (Song et al., 2021a), DPM-Solver (Lu et al., 2022a;b), PNDM (Liu et al., 2022a), and DEIS (Zhang & Chen, 2023). Specifically, Wang et al. (2025) proposed the combination of past and future

*Table 4.* Training settings for learning rate (LR), loss function (Loss), number of ground truth trajectories (Trajectory), and tolerance $\tau$ (Tolerance) when applying PAS to correct DDIM (Song et al., 2021a) and iPNDM (Liu et al., 2022a; Zhang & Chen, 2023) solvers across various datasets and pre-trained models.

| Method (+ PAS) | LR | Loss | Trajectory | Tolerance |
|---|---|---|---|---|
| **CIFAR10 32×32** (Krizhevsky et al., 2009) | | | | |
| DDIM (Song et al., 2021a) | $10^{-2}$ | $L_1$ | 10k | $10^{-2}$ |
| iPNDM (Zhang & Chen, 2023) | 1 | $L_1$ | 5k | $10^{-4}$ |
| **FFHQ 64×64** (Karras et al., 2019) | | | | |
| DDIM (Song et al., 2021a) | $10^{-2}$ | $L_1$ | 10k | $10^{-2}$ |
| iPNDM (Zhang & Chen, 2023) | $10^{-3}$ | $L_1$ | 5k | $10^{-4}$ |
| **ImageNet 64×64** (Deng et al., 2009) | | | | |
| DDIM (Song et al., 2021a) | $10^{-2}$ | $L_1$ | 10k | $10^{-2}$ |
| iPNDM (Zhang & Chen, 2023) | $10^{-3}$ | $L_2$ | 5k | $10^{-4}$ |
| **LSUN Bedroom 256×256** (Yu et al., 2015) | | | | |
| DDIM (Song et al., 2021a) | $10^{-2}$ | $L_1$ | 5k | $10^{-2}$ |
| iPNDM (Zhang & Chen, 2023) | $10^{-2}$ | $L_1$ | 5k | $10^{-4}$ |
| **Stable Diffusion 512×512** (Rombach et al., 2022) | | | | |
| DDIM (Song et al., 2021a) | 10 | $L_1$ | 5k | $10^{-2}$ |

scores to efficiently utilize information and thus reduce discretization error. Ma et al. (2024) and Wimbauer et al. (2024) reduced the computational load of neural networks by caching their low-level features. Shih et al. (2023) suggested utilizing more computational resources and implementing parallelized sampling processes to shorten sampling times. *Orthogonal to these studies*, the proposed PAS method introduces a new orthogonal axis for accelerated sampling in DPMs, which can be further integrated with these approaches to enhance the sampling efficiency of existing fast solvers.

## B. Training Details and Discussion

In this section, we provide training details regarding the PAS correction for different solvers (including DDIM (Song et al., 2021a) and iPNDM (Liu et al., 2022a; Zhang & Chen, 2023)) across various datasets. Unless mentioned in the ablation experiments or special notes, all experimental settings related to training are based on what is described in this section. First, we outline some common experimental settings: we use Heun's 2nd solver from EDM (Karras et al., 2022) with 100 NFE to generate ground truth trajectories. We uniformly apply four orthogonal unit basis vectors (where $\mathbf{u}_1 = d_{t_{i+1}}/\|d_{t_{i+1}}\|_2$) to correct the sampling directions. Notably, for the PCA process in Equation (10), we utilize the `torch.pca_lowrank` function to obtain the basis vectors, as it offers a faster computation speed compared to `torch.svd`. Second, other hyperparameters such as learning rate, loss function, the number of ground truth trajectories, and tolerance $\tau$ are specified in Table 4.

*Table 5.* Sample quality measured by Fréchet Inception Distance (FID↓) on CIFAR10 32×32 (Krizhevsky et al., 2009), FFHQ 64×64 (Karras et al., 2019) datasets, varying the number of function evaluations (NFE) from 4 to 10. "\" indicates missing data due to the inherent characteristics of the algorithm.

| Method | NFE | | | | | | |
|---|---|---|---|---|---|---|---|
| | 4 | 5 | 6 | 7 | 8 | 9 | 10 |
| **CIFAR10 32×32** (Krizhevsky et al., 2009) | | | | | | | |
| DDIM (Song et al., 2021a) | 66.76 | 49.68 | 35.63 | 27.93 | 22.32 | 18.43 | 15.69 |
| DDIM + PAS (**Ours**) | 41.14 | 17.13 | 12.11 | 11.77 | 7.07 | 5.56 | 4.37 |
| Heun's 2nd (Karras et al., 2022) | 319.87 | \ | 99.74 | \ | 38.06 | \ | 15.93 |
| DPM-Solver-2 (Lu et al., 2022a) | 145.98 | \ | 60.00 | \ | 10.30 | \ | 5.01 |
| DPM-Solver++(3M) (Lu et al., 2022b) | 50.39 | 31.65 | 17.89 | 11.30 | 8.30 | 6.45 | 5.16 |
| DEIS-tAB3 (Zhang & Chen, 2023) | 47.13 | 17.65 | 11.84 | 10.89 | 6.82 | 6.21 | 5.64 |
| UniPC(3M) (Zhao et al., 2023) | 49.79 | 31.44 | 17.74 | 11.24 | 8.42 | 6.69 | 5.31 |
| iPNDM (Zhang & Chen, 2023) | 29.49 | 16.55 | 9.74 | 6.92 | 5.23 | 4.33 | 3.69 |
| iPNDM + PAS (**Ours**) | **27.59** | **13.61** | **7.47** | **5.59** | **3.87** | **3.17** | **2.84** |
| **FFHQ 64×64** (Karras et al., 2019) | | | | | | | |
| DDIM (Song et al., 2021a) | 57.48 | 43.92 | 35.21 | 28.86 | 24.38 | 21.01 | 18.37 |
| DDIM + PAS (**Ours**) | 39.09 | 29.07 | 17.63 | 12.47 | 8.16 | 8.26 | 5.61 |
| Heun's 2nd (Karras et al., 2022) | 344.87 | \ | 142.39 | \ | 57.21 | \ | 29.54 |
| DPM-Solver-2 (Lu et al., 2022a) | 238.57 | \ | 83.17 | \ | 22.84 | \ | 9.46 |
| DPM-Solver++(3M) (Lu et al., 2022b) | 39.50 | 23.50 | 14.93 | 11.04 | 9.58 | 8.36 | 6.96 |
| DEIS-tAB3 (Zhang & Chen, 2023) | 35.34 | 19.47 | 11.61 | 11.70 | 8.64 | 7.72 | 7.07 |
| UniPC(3M) (Zhao et al., 2023) | 38.60 | 22.82 | 14.30 | 10.90 | 10.07 | 9.00 | 7.39 |
| iPNDM (Zhang & Chen, 2023) | **29.07** | 17.26 | 11.31 | 8.56 | 6.82 | 5.71 | 4.95 |
| iPNDM + PAS (**Ours**) | 41.89 | **15.89** | **10.29** | **7.59** | **5.85** | **4.88** | **4.28** |

Regarding the aforementioned hyperparameter settings, we conducted extensive ablation experiments in Section 4.3 and Appendix C.2 to elucidate the rationale behind these choices. Furthermore, we note that the impact of hyperparameter settings on the correction of DDIM is not a critical factor, as DDIM exhibits substantial truncation error; regardless of how hyperparameters are configured, PAS significantly enhances the sampling quality of DDIM. In contrast, since the iPNDM solver's sampling quality is already relatively high, certain hyperparameters need to be adjusted when using PAS to achieve better FID scores. Nevertheless, due to the extremely low training cost of PAS (requiring only 0∼2 minutes on a single NVIDIA A100 GPU for the CIFAR10 and 10∼20 minutes for larger datasets at a resolution of 256), we can *easily* conduct hyperparameter searches. Coincidentally, the final training loss can serve as a reference for assessing the effectiveness of hyperparameter choices.

Regarding hyperparameter search recommendations, for solvers with significant truncation errors (*e.g.*, DDIM), we suggest using a learning rate of $10^{-2}$, the $L_1$ loss function, 5k ground truth trajectories, and a tolerance $\tau$ of $10^{-2}$, which generally applies to all datasets. Conversely, for solvers with smaller truncation errors (*e.g.*, iPNDM (Liu et al., 2022a; Zhang & Chen, 2023)), we recommend conducting a learning rate search from $10^{-4}$ to 10 for different datasets, while fixing the $L_1$ loss function, using 5k ground truth trajectories, setting the tolerance $\tau$ to $10^{-4}$.

It is important to emphasize that the mainstream evaluation metric for the quality of generated samples is FID; however, there is no corresponding FID loss function. Therefore, when training coordinates using the $L_1$ or $L_2$ loss functions,

even if the FID score does not improve during the correction of the iPNDM solver, the $L_1$ and $L_2$ metrics show improvement, as demonstrated in Table 11. This further corroborates the effectiveness of the proposed PAS as a plug-and-play correction algorithm.

## C. Additional Experiment Results

In this section, we present additional experimental results on NFE, corrected time points, the order of iPNDM, ablation studies, and visualization studies. Except for the ablation experiments and specific clarifications, the experimental setup and training details are consistent with Section 4 and Appendix B.

### C.1. Additional Results on NFE and Corrected Time Points

In this section, we first extend the FID results on the CIFAR10 32×32 (Krizhevsky et al., 2009) and FFHQ 64×64 (Karras et al., 2019) datasets for more values of NFE $\in \{4, 5, 6, 7, 8, 9, 10\}$. The results of PAS correcting DDIM (Song et al., 2021a) and iPNDM (Liu et al., 2022a; Zhang & Chen, 2023) with the order of 3 are shown in Table 5 (for more results regarding the order of iPNDM, see Appendix C.3). The results indicate that PAS can significantly improve the sampling quality of DDIM and iPNDM across different NFE.

Additionally, in Table 6, we present the time points corrected by PAS for the DDIM and iPNDM solvers across various datasets, which correspond to Tables 2 and 3. From

*Table 6.* The list of numbers denotes all the time points requiring correction from adaptive search for the DDIM (Song et al., 2021a) and iPNDM (Liu et al., 2022a; Zhang & Chen, 2023) solvers, with each element $i$ ranging from NFE ($N$) to 1.

| Method | NFE | | | |
|---|---|---|---|---|
| | 5 | 6 | 8 | 10 |
| **CIFAR10 32×32** (Krizhevsky et al., 2009) | | | | |
| DDIM + PAS | 3,1 | 4,2,1 | 5,3,2 | 6,4,2 |
| iPNDM + PAS | 2 | 3 | 3,1 | 4,2 |
| **FFHQ 64×64** (Karras et al., 2019) | | | | |
| DDIM + PAS | 3,2,1 | 4,3,1 | 5,4,2,1 | 7,5,3,2 |
| iPNDM + PAS | 3 | 3,1 | 4,2 | 4,2 |
| **ImageNet 64×64** (Deng et al., 2009) | | | | |
| DDIM + PAS | 3,2,1 | 4,3,1 | 5,4,2,1 | 7,5,4,2,1 |
| iPNDM + PAS | 3 | 3 | 4 | 5 |
| **LSUN Bedroom 256×256** (Yu et al., 2015) | | | | |
| DDIM + PAS | 4,3,1 | 5,4,2 | 6,5,3,2 | 8,7,5,4 |
| iPNDM + PAS | 4,2 | 5,3,2,1 | 6,4,3,1 | 8,6,4,3 |
| **Stable Diffusion 512×512** (Rombach et al., 2022) | | | | |
| DDIM + PAS | 3,2,1 | 2,1 | 2,1 | 1 |

Table 6, it can be observed that the DDIM, which has a large truncation error, requires correction of more sampling steps, while the iPNDM, which exhibits a smaller truncation error, requires relatively fewer correction steps, which *aligns with our intuition*. Overall, PAS only needs to correct 1∼5 time points, which corresponds to requiring only 4∼20 learnable parameters, to significantly enhance the sampling quality of the baseline solvers. This validates the effectiveness of PAS as an acceleration algorithm with minimal training costs.

### C.2. Additional Ablation Study Results

In this section, we first supplement additional ablation experiments concerning the adaptive search. Subsequently, we provide further ablation experiments on learning rate, tolerance $\tau$, and solvers for trajectory generation.

**More results on adaptive search.** Regarding the adaptive search strategy, we first supplement the specific FID values corresponding to Figure 6a on the CIFAR10 32×32 (Krizhevsky et al., 2009) dataset, as shown in Table 7. Additionally, in Table 7, we also provide experimental results on the FFHQ 64×64 (Karras et al., 2019) dataset, using PAS alongside PAS without the adaptive search strategy (PAS(-AS)) to correct DDIM (Song et al., 2021a). Consistent with the results in Figure 6a, when the adaptive search strategy is not employed—specifically when PCA-based sampling correction is applied at each time step—the sampling quality is inferior to that of the baseline DDIM. Furthermore, the absence of the adaptive search strategy results in increased computational time, as PCA-based sampling correction is required at each step. In Section 3.3, we described the motivation and process of the proposed

*Table 7.* Ablation study regarding adaptive search conducted on the CIFAR10 32×32 (Krizhevsky et al., 2009) and FFHQ 64×64 (Karras et al., 2019) datasets, employing PAS alongside PAS without adaptive search (PAS (-AS)) to correct DDIM (Song et al., 2021a). We report the Fréchet Inception Distance (FID↓) score, varying the number of function evaluations (NFE).

| Method | NFE | | | |
|---|---|---|---|---|
| | 5 | 6 | 8 | 10 |
| **CIFAR10 32×32** (Krizhevsky et al., 2009) | | | | |
| DDIM (Song et al., 2021a) | 49.68 | 35.63 | 22.32 | 15.69 |
| DDIM + PAS (-AS) | 132.12 | 139.48 | 143.54 | 120.32 |
| DDIM + PAS | **17.13** | **12.11** | **7.07** | **4.37** |
| **FFHQ 64×64** (Karras et al., 2019) | | | | |
| DDIM (Song et al., 2021a) | 43.92 | 35.21 | 24.38 | 18.37 |
| DDIM + PAS (-AS) | 78.10 | 98.84 | 98.67 | 93.62 |
| DDIM + PAS | **29.07** | **17.63** | **8.16** | **5.61** |

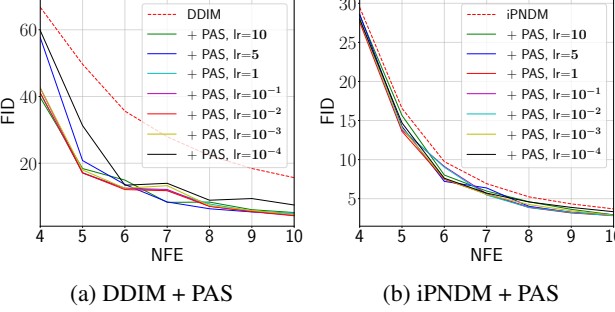

(a) DDIM + PAS      (b) iPNDM + PAS

*Figure 7.* Ablation study regarding learning rate conducted on the CIFAR10 32×32 (Krizhevsky et al., 2009), utilizing PAS to correct DDIM (Song et al., 2021a) and iPNDM (Liu et al., 2022a; Zhang & Chen, 2023) solvers. We report the Fréchet Inception Distance (FID↓) score, varying the number of function evaluations (NFE).

adaptive search strategy. We analyzed the ground truth trajectory transitioning from a straight line to a curve, and ultimately back to a straight line. The adaptive search is designed to correct the errors in the large curvature regions of the sampling trajectory. In the straight linear segments, the errors introduced by the existing fast solvers (*e.g.*, DDIM) are negligible, and the application of PCA-based sampling correction does not provide any further error adjustment. Instead, it introduces biases on other basis vectors, leading to a decline in sampling quality. Therefore, it is essential to combine the adaptive search strategy with PCA-based sampling correction, specifically to correct truncation errors in regions of high curvature in the sampling trajectory. Combining Figure 6a and Table 7, further validates the necessity and effectiveness of the proposed overall algorithm, PCA-based Adaptive Search (PAS).

**Learning rate.** In Figure 7, we present the ablation results of the PAS correcting DDIM (Song et al., 2021a) and iPNDM (Liu et al., 2022a; Zhang & Chen, 2023) with the order of 3 on the CIFAR10 32×32 (Krizhevsky et al., 2009)

*Table 8.* Ablation study regarding tolerance $\tau$ conducted on the CIFAR10 32×32 (Krizhevsky et al., 2009), utilizing PAS to correct DDIM (Song et al., 2021a) and iPNDM (Liu et al., 2022a; Zhang & Chen, 2023) solvers. We report the Fréchet Inception Distance (FID↓) score, varying the number of function evaluations (NFE).

| Method | $\tau$ | NFE | | | |
|---|---|---|---|---|---|
| | | 5 | 6 | 8 | 10 |
| **CIFAR10 32×32** (Krizhevsky et al., 2009) | | | | | |
| DDIM (Song et al., 2021a) | \ | 49.68 | 35.63 | 22.32 | 15.69 |
| DDIM + PAS | $10^{-1}$ | 49.68 | 35.63 | 22.32 | 15.69 |
| DDIM + PAS | $10^{-2}$ | 17.13 | 12.11 | 7.07 | 4.37 |
| DDIM + PAS | $10^{-3}$ | 17.13 | 12.11 | 7.07 | 4.37 |
| DDIM + PAS | $10^{-4}$ | 17.13 | 12.11 | 7.07 | 4.37 |
| iPNDM (Zhang & Chen, 2023) | \ | 16.55 | 9.74 | 5.23 | 3.69 |
| iPNDM + PAS | $10^{-2}$ | 13.61 | 9.74 | 5.23 | 3.69 |
| iPNDM + PAS | $10^{-3}$ | 13.61 | 7.47 | 3.87 | 2.91 |
| iPNDM + PAS | $10^{-4}$ | 13.61 | 7.47 | 3.87 | 2.84 |

*Table 9.* Ablation study regarding solvers for generating ground truth trajectories, including Heun's 2nd (Heun) (Karras et al., 2022), DDIM (Song et al., 2021a), and DPM-Solver-2 (DPM) (Lu et al., 2022a), conducted on the CIFAR10 32×32 (Krizhevsky et al., 2009) and FFHQ 64×64 (Karras et al., 2019). We report the Fréchet Inception Distance (FID↓) score, employing PAS to correct DDIM (Song et al., 2021a), varying the NFE.

| Method | Solver | NFE | | | |
|---|---|---|---|---|---|
| | | 5 | 6 | 8 | 10 |
| **CIFAR10 32×32** (Krizhevsky et al., 2009) | | | | | |
| DDIM (Song et al., 2021a) | \ | 49.68 | 35.63 | 22.32 | 15.69 |
| DDIM + PAS | Heun | 17.13 | 12.11 | **7.07** | **4.37** |
| DDIM + PAS | DDIM | **17.10** | 12.44 | 6.97 | 4.87 |
| DDIM + PAS | DPM | 17.12 | **12.10** | 7.10 | 4.40 |
| **FFHQ 64×64** (Karras et al., 2019) | | | | | |
| DDIM (Song et al., 2021a) | \ | 43.92 | 35.21 | 24.38 | 18.37 |
| DDIM + PAS | Heun | **29.07** | 17.63 | 8.16 | **5.61** |
| DDIM + PAS | DDIM | 30.49 | **15.26** | 8.65 | 6.37 |
| DDIM + PAS | DPM | 29.11 | 17.58 | **8.12** | 5.64 |

dataset. We varied the learning rate from $10^{-4}$ to 10. The results demonstrate that, regardless of the learning rate setting, PAS significantly improves the sampling quality of both DDIM and iPNDM on the CIFAR10, and further exploration exhibits slightly better sampling performance.

**Tolerance $\tau$.** In Section 3.3, we designed an adaptive search strategy to enhance the overall performance of PCA-based sampling correction. This adaptive search strategy relies on the condition $\mathcal{L}_2 - (\mathcal{L}_1 + \tau) > 0$. Therefore, we further investigate the impact of the tolerance $\tau$ on the adaptive search strategy. It is important to emphasize that the adaptive search strategy is aimed at correcting the sections of the sampling trajectory with large curvature, while the tolerance $\tau$ serves as the criterion for determining whether the current sampling state has reached a region of large curvature in the trajectory. Consequently, the tolerance $\tau$ is initially treated as a hyperparameter to indicate the time

*Table 10.* Sample quality measured by Fréchet Inception Distance (FID↓) on the LSUN Bedroom 256×256 (Yu et al., 2015) dataset and Stable Diffusion (Rombach et al., 2022), varying the order of iPNDM (Liu et al., 2022a; Zhang & Chen, 2023).

| Method | Order | NFE | | | |
|---|---|---|---|---|---|
| | | 5 | 6 | 8 | 10 |
| **LSUN Bedroom 256×256** (Yu et al., 2015) | | | | | |
| iPNDM (Zhang & Chen, 2023) | 1 | 34.34 | 25.25 | 15.71 | 11.42 |
| iPNDM (Zhang & Chen, 2023) | 2 | 18.15 | 12.90 | 7.98 | 6.17 |
| iPNDM (Zhang & Chen, 2023) | 3 | 16.57 | 10.83 | 6.18 | 4.92 |
| iPNDM (Zhang & Chen, 2023) | 4 | 26.65 | 20.72 | 11.77 | 5.56 |
| iPNDM + PAS | 2 | **15.48** | **10.24** | 6.67 | 5.14 |
| iPNDM + PAS | 3 | 18.59 | 12.06 | **5.92** | **4.84** |
| **Stable Diffusion 512×512** (Rombach et al., 2022) | | | | | |
| iPNDM (Zhang & Chen, 2023) | 3 | 22.31 | 17.21 | **13.60** | **13.71** |
| iPNDM (Zhang & Chen, 2023) | 4 | 28.12 | 25.69 | 20.68 | 16.45 |
| DDIM (Song et al., 2021a) | \ | 23.42 | 20.08 | 17.72 | 16.56 |
| DDIM + PAS | \ | **17.70** | **15.93** | 14.74 | 14.23 |

point at which correction should commence, while subsequent time points have their tolerance $\tau$ fixed at $10^{-4}$. We adjusted the tolerance $\tau$ for the initial correction point from $10^{-1}$ to $10^{-4}$, and the experimental results on the CIFAR10 32×32 (Krizhevsky et al., 2009) dataset are presented in Table 8. The experimental findings indicate that PAS is not sensitive to the configuration of the hyperparameter tolerance $\tau$. PAS consistently demonstrates a significant improvement in the sampling quality of DDIM (Song et al., 2021a) and iPNDM (Liu et al., 2022a; Zhang & Chen, 2023) solvers with tolerance $\tau$ ranging from $10^{-2}$ to $10^{-4}$. Lastly, we recommend setting the tolerance $\tau$ to $10^{-2}$ for solvers with substantial truncation error (*e.g.*, DDIM), while for solvers with relatively smaller truncation error (*e.g.*, iPNDM), the tolerance $\tau$ should be set to $10^{-4}$.

**Solvers for trajectory generation** We investigated the impact of solver selection for generating ground truth trajectories on the performance of PAS using the CIFAR10 32×32 (Krizhevsky et al., 2009) and FFHQ 64×64 (Karras et al., 2019) datasets. We employed Heun's 2nd (Karras et al., 2022), DDIM (Song et al., 2021a), and DPM-Solver-2 (Lu et al., 2022a) solvers with 100 NFE to generate 10k trajectories for training the PAS to correct the DDIM solver, as shown in Table 9. Our findings indicate that the choice of solver for generating ground truth trajectories has negligible impact on the performance of PAS. This suggests that regardless of the solver used, a sufficient number of NFE (*e.g.*, 100 NFE) enables the solving process to approximate the ground truth trajectories closely. Therefore, we simply fixed Heun's 2nd solver for all experiments.

### C.3. Additional Results on the Order of iPNDM

In this section, we discuss why we mainly chose to apply PAS correcting iPNDM with the order of 3 in Table 2. First,

*Table 11.* Sample quality measured by Fréchet Inception Distance (FID↓), $L_2$ (MSE)↓, and $L_1$↓ metrics on the CIFAR10 32×32 (Krizhevsky et al., 2009) dataset, varying the order of iPNDM (Liu et al., 2022a; Zhang & Chen, 2023). The $L_2$ (MSE) and $L_1$ metrics were evaluated against Heun's 2nd solver (Karras et al., 2022) with 100 NFE, using 50k samples.

| Method | Order | Metrics | NFE | | | | | | |
|---|---|---|---|---|---|---|---|---|---|
| | | | 4 | 5 | 6 | 7 | 8 | 9 | 10 |
| **CIFAR10 32×32** (Krizhevsky et al., 2009), Fréchet Inception Distance (FID↓) metric | | | | | | | | | |
| iPNDM (Zhang & Chen, 2023) | 1 | FID↓ | 66.76 | 49.68 | 35.63 | 27.93 | 22.32 | 18.43 | 15.69 |
| iPNDM + PAS (**Ours**) | 1 | FID↓ | 41.14 | 17.13 | 12.11 | 11.77 | 7.07 | 5.56 | 4.37 |
| iPNDM (Zhang & Chen, 2023) | 2 | FID↓ | 39.02 | 25.24 | 16.19 | 11.85 | 9.08 | 7.39 | 6.18 |
| iPNDM + PAS (**Ours**) | 2 | FID↓ | 33.54 | 16.59 | 9.77 | 5.87 | 4.51 | 3.55 | 3.01 |
| iPNDM (Zhang & Chen, 2023) | 3 | FID↓ | 29.49 | 16.55 | 9.74 | 6.92 | 5.23 | 4.33 | 3.69 |
| iPNDM + PAS (**Ours**) | 3 | FID↓ | 27.59 | 13.61 | 7.47 | 5.59 | 3.87 | 3.17 | 2.84 |
| iPNDM (Zhang & Chen, 2023) | 4 | FID↓ | 24.82 | 13.58 | 7.05 | 5.08 | 3.69 | 3.17 | 2.77 |
| iPNDM + PAS (**Ours**) | 4 | FID↓ | 25.79 | 13.74 | 7.95 | 5.89 | 4.66 | 3.42 | 2.97 |
| **CIFAR10 32×32** (Krizhevsky et al., 2009), $L_2$ (MSE) and $L_1$ metrics | | | | | | | | | |
| iPNDM (Zhang & Chen, 2023) | 4 | $L_2$ (MSE)↓ | 0.027 | 0.016 | 0.009 | 0.006 | 0.004 | 0.003 | 0.002 |
| iPNDM + PAS (**Ours**) | 4 | $L_2$ (MSE)↓ | 0.020 | 0.014 | 0.009 | 0.006 | 0.004 | 0.003 | 0.002 |
| iPNDM (Zhang & Chen, 2023) | 4 | $L_1$↓ | 0.126 | 0.089 | 0.063 | 0.047 | 0.037 | 0.030 | 0.025 |
| iPNDM + PAS (**Ours**) | 4 | $L_1$↓ | 0.100 | 0.078 | 0.059 | 0.047 | 0.037 | 0.031 | 0.026 |

we evaluated the sampling quality of iPNDM by adjusting its order on the CIFAR10 32×32 (Krizhevsky et al., 2009), LSUN Bedroom 256×256 (Yu et al., 2015) datasets, and Stable Diffusion (Rombach et al., 2022), as shown in Tables 10 and 11. We found that increasing the order of iPNDM does not always enhance sampling quality. Particularly on high-resolution datasets, optimal sampling quality is often not achieved at iPNDM with the order of 4; in comparison, iPNDM with order 3 demonstrates better average performance. Therefore, we mainly selected PAS to correct the iPNDM with the order of 3 in Table 2.

Furthermore, on low-resolution datasets, such as CIFAR10 32×32, iPNDM with the order of 4 displays the best performance. Consequently, we employed PAS to correct its truncation error; however, we observed no improvement in its FID score. Nevertheless, the iPNDM corrected by PAS with the order of 4 performs better in terms of $L_1$ and $L_2$ metrics, as indicated in Table 11. This phenomenon may be attributed to the fact that PAS uses $L_1$ or $L_2$ loss functions during training, and when the solver's sampling quality is already relatively satisfactory, *there is not always consistency between the $L_1$, $L_2$ metrics and the FID score*.

### C.4. Additional Visualize Study Results

We present additional visual sampling results using Stable Diffusion v1.4 (Rombach et al., 2022), as shown in Figure 8. Furthermore, more visual results on the CIFAR10 32×32 (Krizhevsky et al., 2009), FFHQ 64×64 (Karras et al., 2019), ImageNet 64×64 (Deng et al., 2009), and LSUN Bedroom 256×256 (Yu et al., 2015) datasets with NFE of 6 and 10 are displayed in Figures 9 to 16. These visual results demonstrate that the samples generated by PAS exhibit *higher quality and richer details* compared to the corresponding baselines.

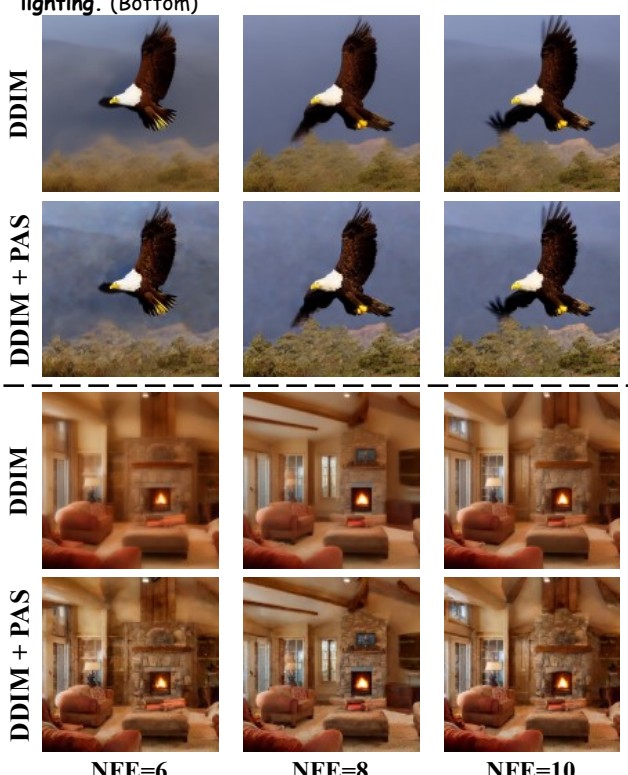

**Text Prompts:**
**Majestic eagle soaring above a mountain landscape.** (Top)
**Warm living room with a crackling fireplace and soft lighting.** (Bottom)

DDIM

DDIM + PAS

DDIM

DDIM + PAS

**NFE=6**   **NFE=8**   **NFE=10**

*Figure 8.* Random samples by DDIM (Song et al., 2021a) with and without the proposed PAS on Stable Diffusion v1.4 (Rombach et al., 2022) with a guidance scale of 7.5.

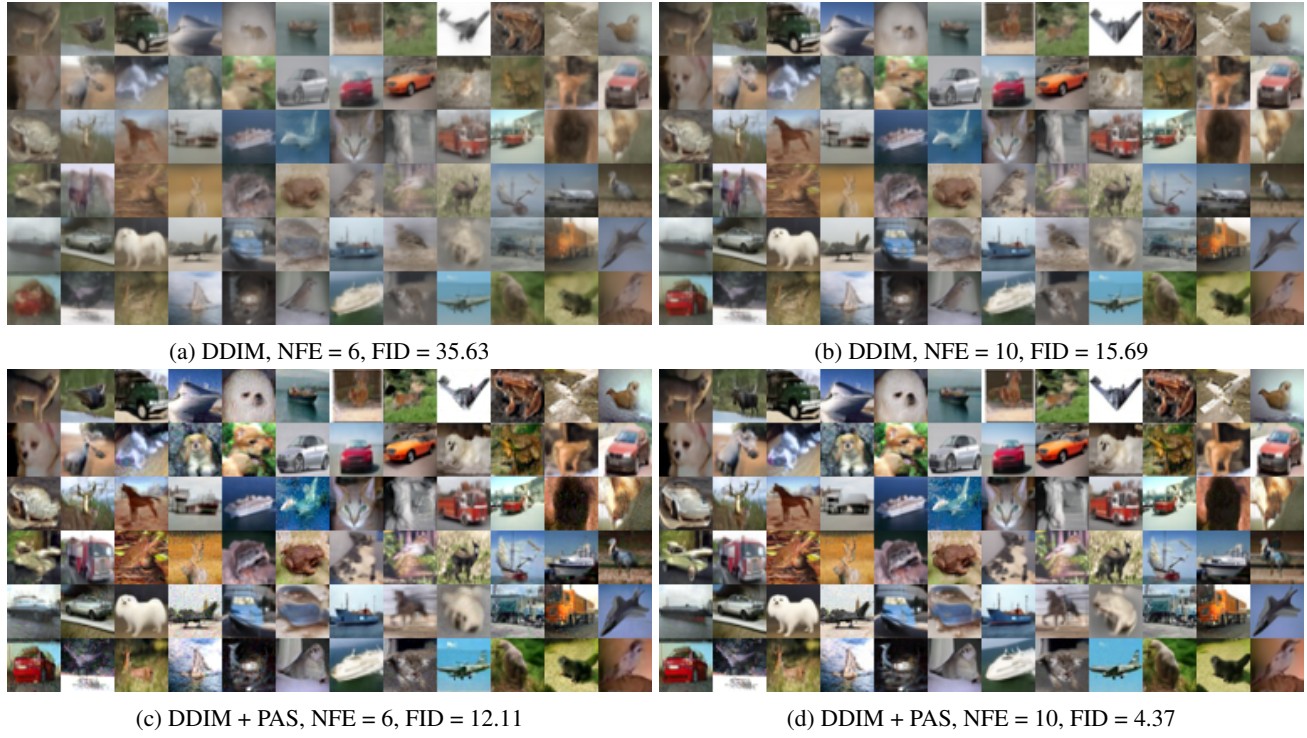

(a) DDIM, NFE = 6, FID = 35.63

(b) DDIM, NFE = 10, FID = 15.69

(c) DDIM + PAS, NFE = 6, FID = 12.11

(d) DDIM + PAS, NFE = 10, FID = 4.37

*Figure 9.* Random samples by DDIM (Song et al., 2021a) with and without the proposed PAS on CIFAR10 32×32.

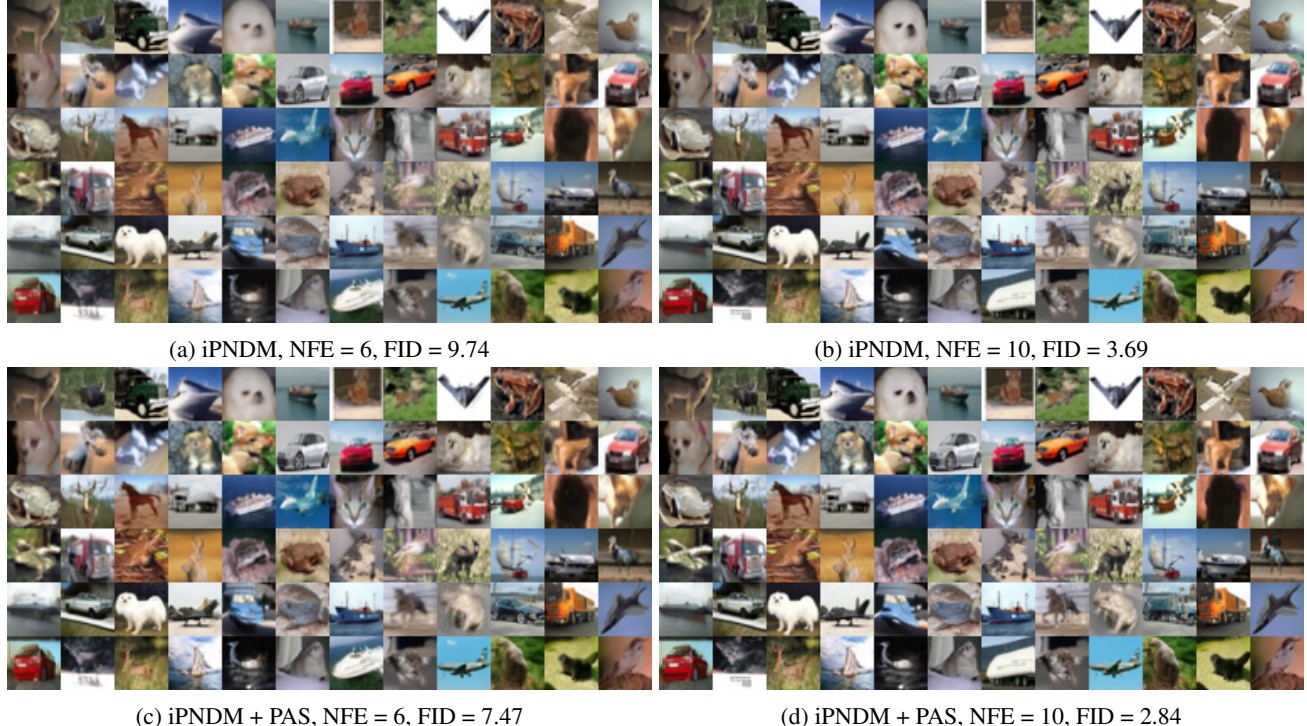

(a) iPNDM, NFE = 6, FID = 9.74

(b) iPNDM, NFE = 10, FID = 3.69

(c) iPNDM + PAS, NFE = 6, FID = 7.47

(d) iPNDM + PAS, NFE = 10, FID = 2.84

*Figure 10.* Random samples by iPNDM (Liu et al., 2022a; Zhang & Chen, 2023) with and without the proposed PAS on CIFAR10 32×32.

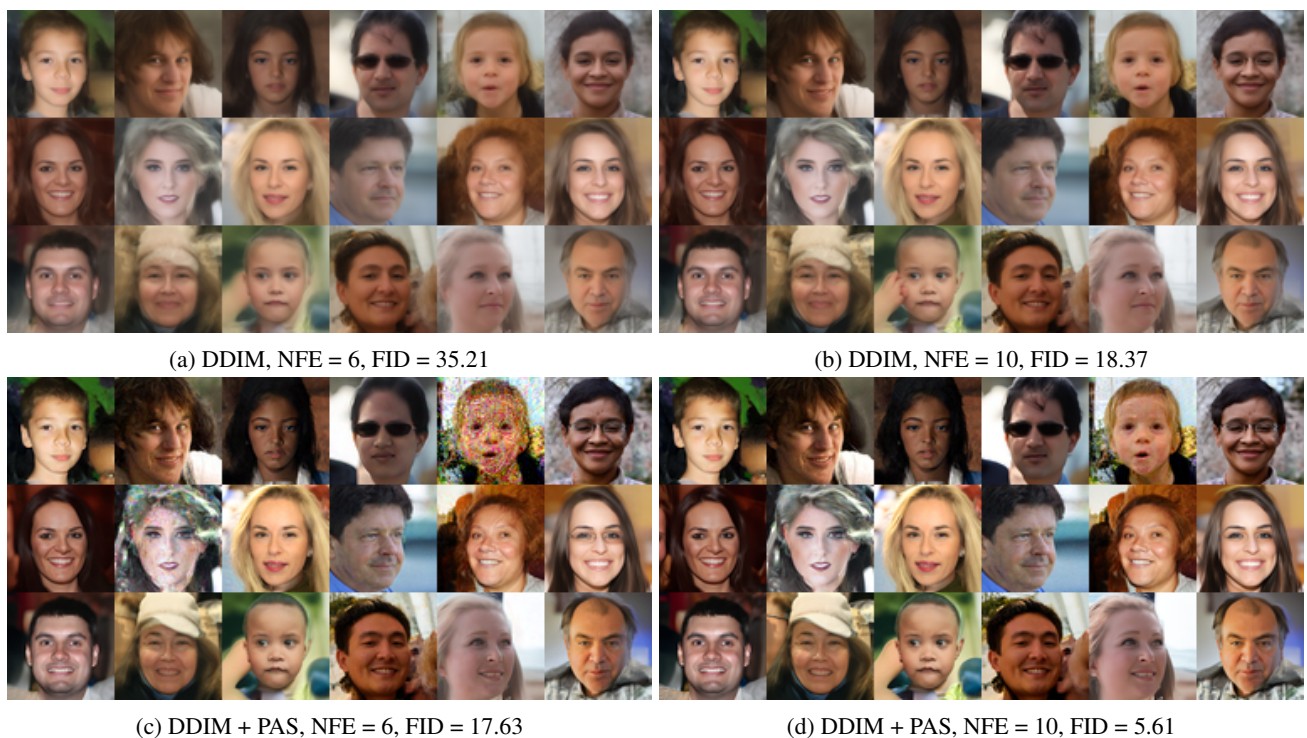

(a) DDIM, NFE = 6, FID = 35.21

(b) DDIM, NFE = 10, FID = 18.37

(c) DDIM + PAS, NFE = 6, FID = 17.63

(d) DDIM + PAS, NFE = 10, FID = 5.61

*Figure 11.* Random samples by DDIM (Song et al., 2021a) with and without the proposed PAS on FFHQ 64×64.

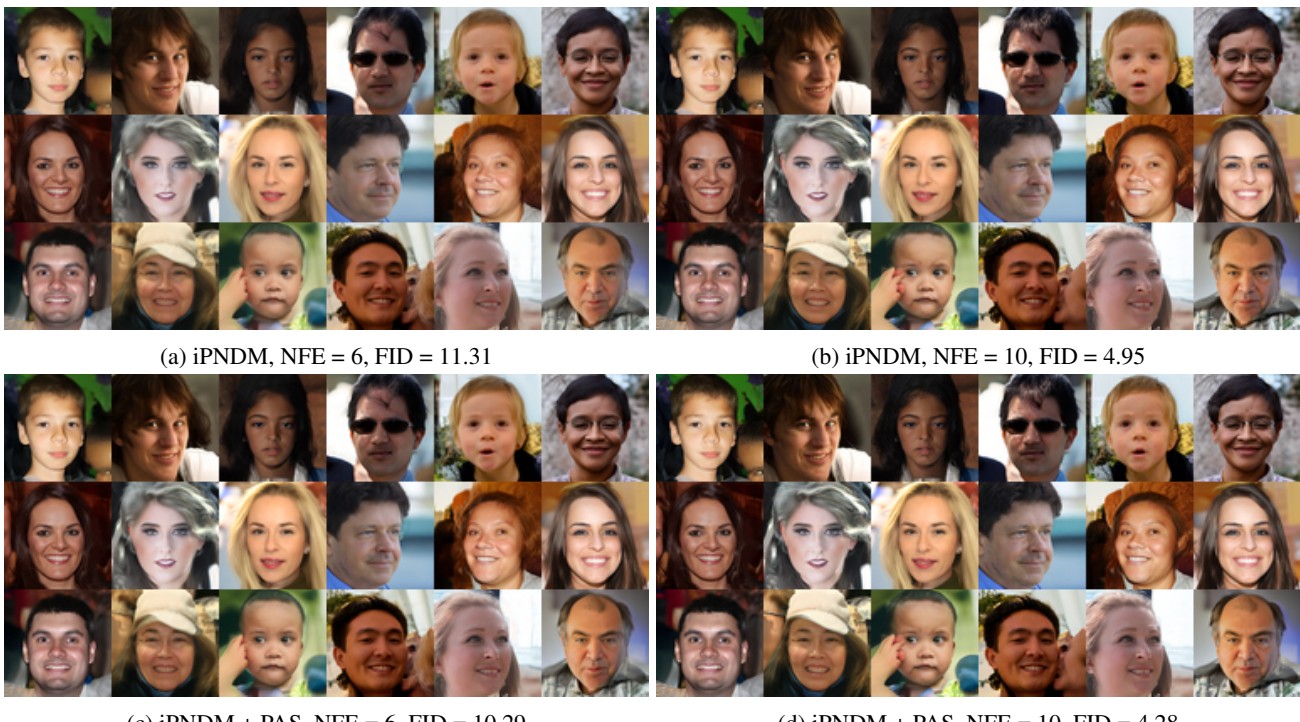

(a) iPNDM, NFE = 6, FID = 11.31

(b) iPNDM, NFE = 10, FID = 4.95

(c) iPNDM + PAS, NFE = 6, FID = 10.29

(d) iPNDM + PAS, NFE = 10, FID = 4.28

*Figure 12.* Random samples by iPNDM (Liu et al., 2022a; Zhang & Chen, 2023) with and without the proposed PAS on FFHQ 64×64.

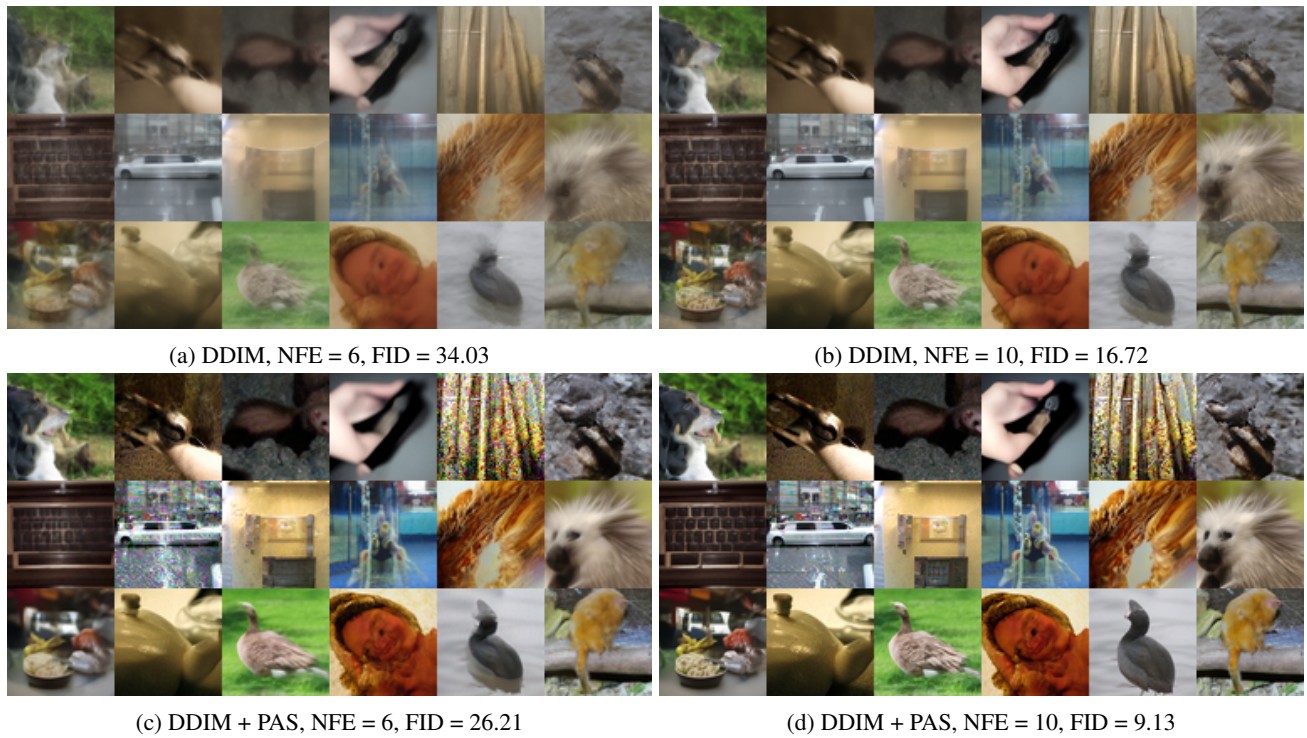

(a) DDIM, NFE = 6, FID = 34.03            (b) DDIM, NFE = 10, FID = 16.72

(c) DDIM + PAS, NFE = 6, FID = 26.21            (d) DDIM + PAS, NFE = 10, FID = 9.13

*Figure 13.* Random samples by DDIM (Song et al., 2021a) with and without the proposed PAS on ImageNet 64×64.

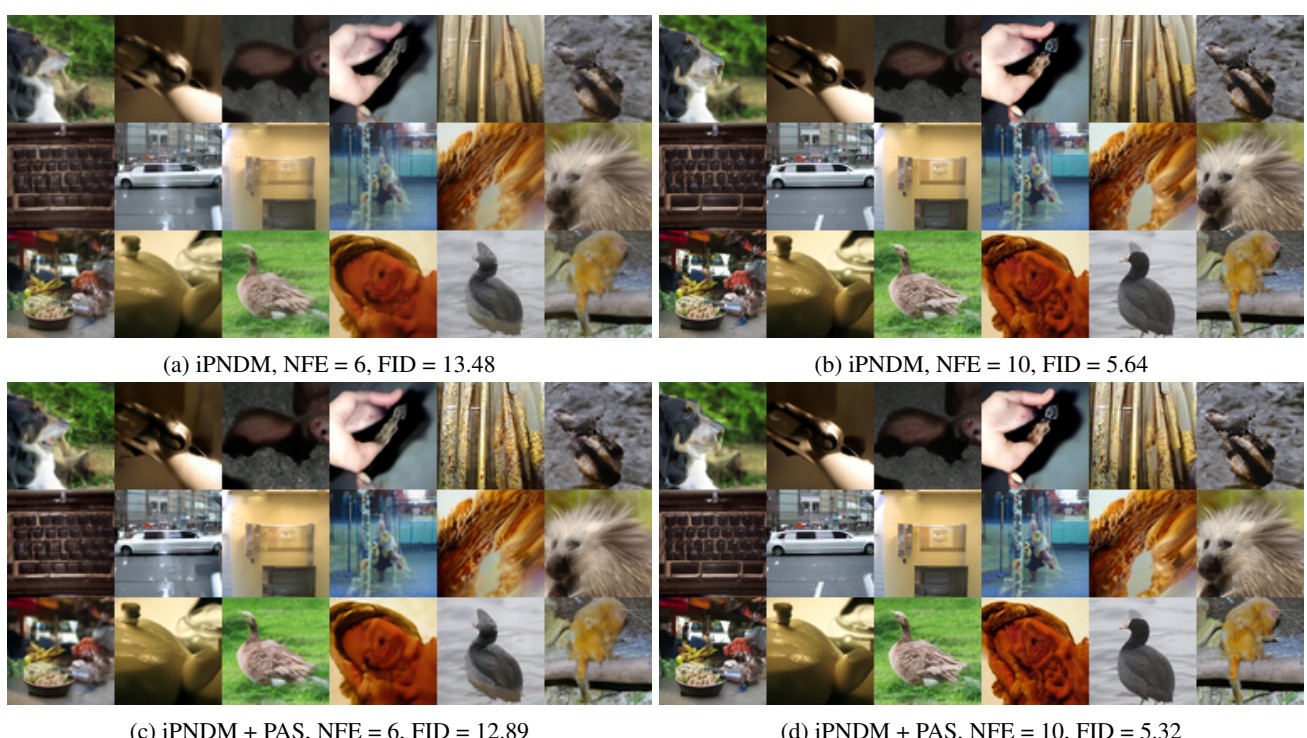

(a) iPNDM, NFE = 6, FID = 13.48            (b) iPNDM, NFE = 10, FID = 5.64

(c) iPNDM + PAS, NFE = 6, FID = 12.89            (d) iPNDM + PAS, NFE = 10, FID = 5.32

*Figure 14.* Random samples by iPNDM (Liu et al., 2022a; Zhang & Chen, 2023) with and without the proposed PAS on ImageNet 64×64.

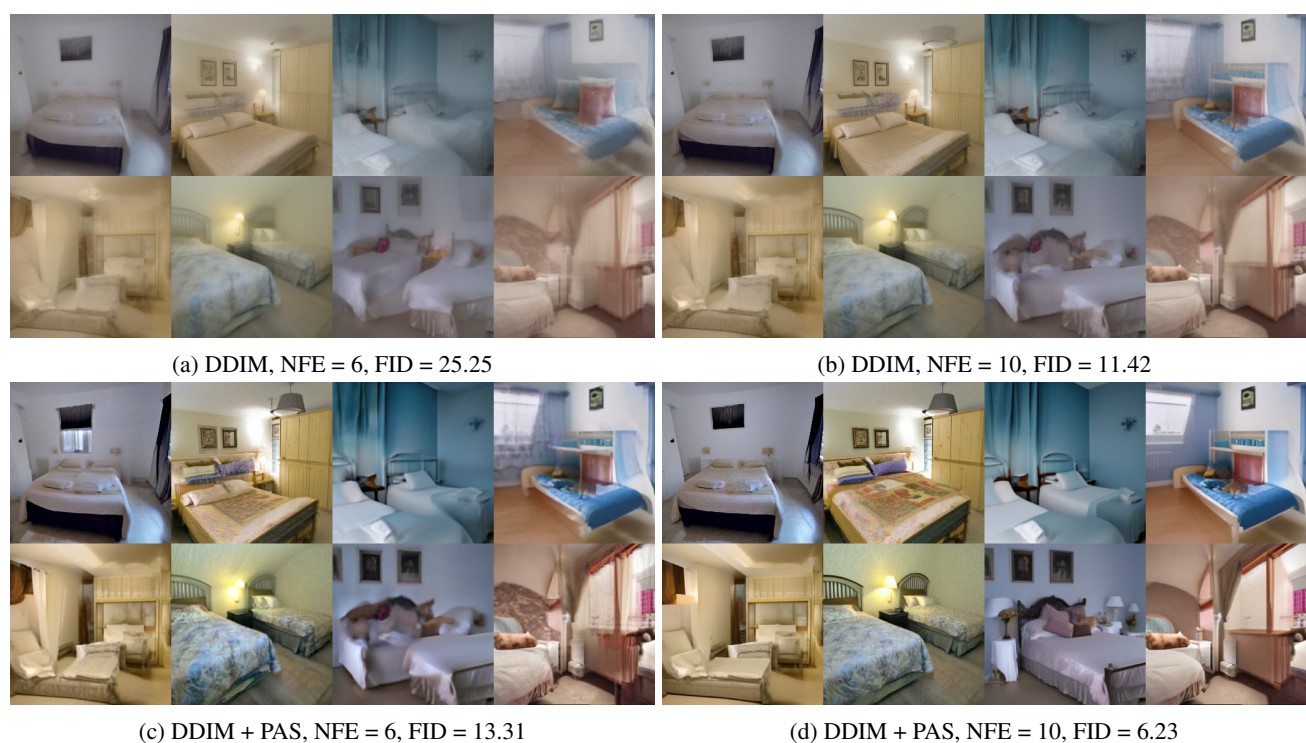

(a) DDIM, NFE = 6, FID = 25.25

(b) DDIM, NFE = 10, FID = 11.42

(c) DDIM + PAS, NFE = 6, FID = 13.31

(d) DDIM + PAS, NFE = 10, FID = 6.23

*Figure 15.* Random samples by DDIM (Song et al., 2021a) with and without the proposed PAS on LSUN Bedroom 256×256.

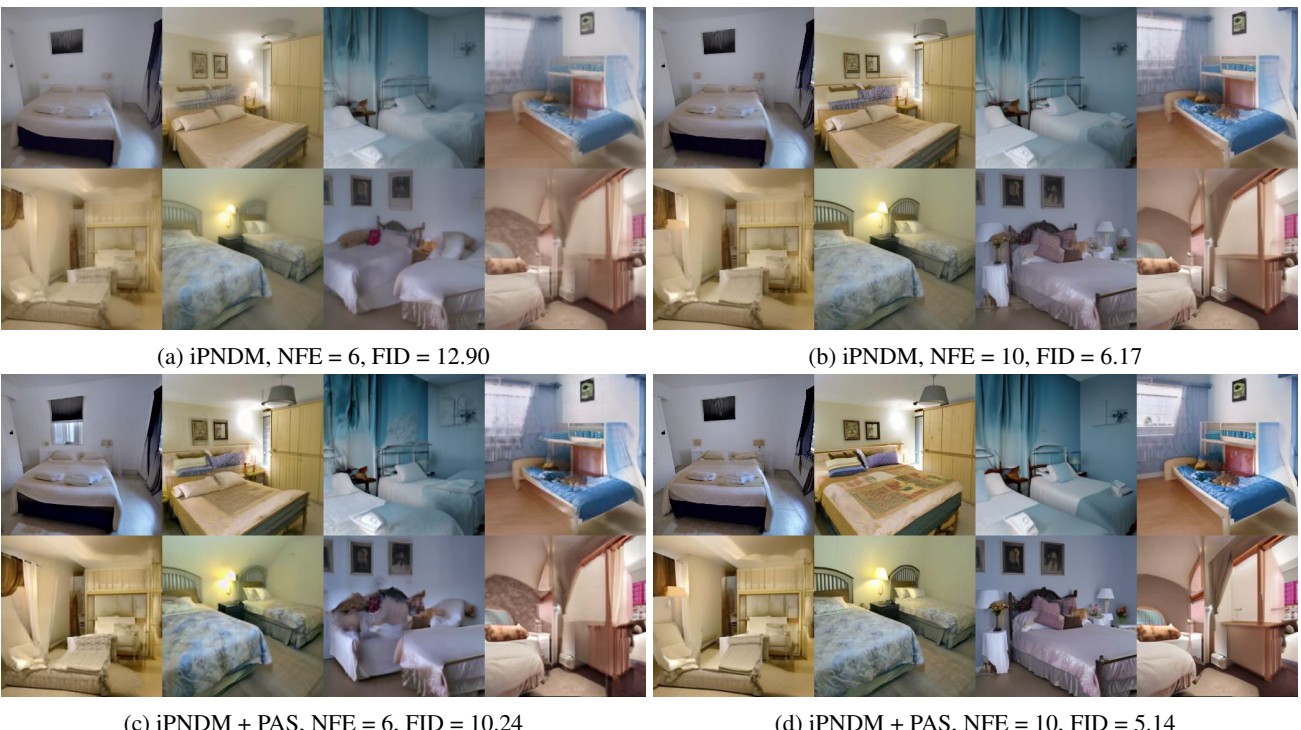

(a) iPNDM, NFE = 6, FID = 12.90

(b) iPNDM, NFE = 10, FID = 6.17

(c) iPNDM + PAS, NFE = 6, FID = 10.24

(d) iPNDM + PAS, NFE = 10, FID = 5.14

*Figure 16.* Random samples by iPNDM (Liu et al., 2022a; Zhang & Chen, 2023) with and without the proposed PAS on LSUN Bedroom 256×256.

