# OpenReview forum: "Diffusion Sampling Correction via Approximately 10 Parameters"
_ICML.cc/2025/Conference — ICML 2025 poster_

### Official Review · Reviewer_rBDg · 2025-03-03

**Overall Recommendation:** 3

**Summary:**

This paper proposes PCA-based Adaptive Search (PAS) to optimize the sampling process of diffusion probabilistic models (DPMs). The key of the method is leveraging Principal Component Analysis to identify a low-dimensional subspace for sampling correction. The method also includes an adaptive search strategy to reduce correction steps. PAS operates in a plug-and-play fashion, enhancing existing fast solvers without significant additional training costs. Experimental results show that PAS significantly improves sampling quality on diverse datasets.

**Claims And Evidence:**

The authors claim that PAS can correct the truncation errors and improve the sample quality. This claim is well-supported by the improvements of FID on multiple datasets.

However, despite the authors claim that PAS can enhance sampling efficiency of existing fast solvers (requiring only a small number of learnable parameters and negligible training time), there is a lack of intuitive comparison with other methods regarding efficiency.

**Essential References Not Discussed:**

The paper cites all major relevant works, including prior fast solvers, trajectory-based corrections, and low-cost training strategies.

**Experimental Designs Or Analyses:**

The experimental design is comprehensive, covering 5 datasets, multiple solvers, and different NFE values (5–10 steps).

Comparisons with other trajectory-based methods (e.g., AMED, GITS) have only been done on DDIM, and it would be useful to compare and analyze combined with other SOTA solvers (e.g., iPNDM).

There is a lack of comparison and analysis about training efficiency between PAS and other methods.

**Methods And Evaluation Criteria:**

The method is well-founded, using PCA on the sampling trajectory to extract low-dimensional basis vectors for sampling correction.

The method is evaluated primarily using FID on standard image generation benchmarks. Comparisons with previously state-of-the-art fast solvers, as well as detailed ablation studies, ensure a comprehensive evaluation.

**Other Comments Or Suggestions:**

None

**Other Strengths And Weaknesses:**

None

**Questions For Authors:**

Have you experimented with alternative dimensionality reduction techniques in place of PCA?

**Relation To Broader Scientific Literature:**

The paper is well-positioned in the broader context of: fast DPM solvers (e.g., DDIM, DPM-Solver-2, DPM-Solver++, iPNDM) and trajectory-based acceleration methods (e.g., AMED, GITS).

It builds on recent findings that sampling trajectories of DPMs lie in a low-dimensional subspace, extending this idea with a novel PCA-based correction approach.

**Theoretical Claims:**

The derivations of PCA-based correction and adaptive search appear correct and well-grounded in numerical methods.

The claim that sampling trajectories lie in a low-dimensional subspace is experimentally validated but could benefit from more theoretical justification.

---

> ### Author Rebuttal · Authors · 2025-03-30
>
> We sincerely appreciate the reviewer's recognition of our work and meticulous review. Please find below our responses to all the questions. We would greatly appreciate it if you could consider increasing the score if you are satisfied with our response.
>
> **Abbreviation: CaE (Claims And Evidence), EDoA (Experimental Designs Or Analyses)**
>
> >***CaE and EDoA2: The study lacks a comparison of training efficiency between PAS and other methods.***
>
> **A**: We provide a comparison of PAS with the previously well-performing AMED and GITS methods, while also introducing the high-cost PD method as a contrast. The comparative experiments, based on sample quality and training cost on CIFAR10, are shown in the table below:
>
> FID↓(A100 hours↓)
>
> | Method\NFE  | 1          | 2          | 4          | 8          |
> | ----------- | ---------- | ---------- | ---------- | ---------- |
> | DDIM+PD[R1] | 9.12(~195) | 4.51(~171) | 3.00(~159) | 2.57(~146) |
>
> | Method\NFE      | 5            | 6            | 8            | 10           |
> | --------------- | ------------ | ------------ | ------------ | ------------ |
> | DDIM+AMED[R2]   | \            | 25.15(~0.08) | 17.03(~0.10) | 11.33(~0.11) |
> | DDIM+GITS[R3]   | 28.05(<0.01) | 21.04(<0.01) | 13.30(<0.01) | 10.37(~0.01) |
> | DDIM+PAS (Ours) | 17.13(~0.01) | 12.11(~0.02) | 7.07(~0.03)  | 4.37(~0.04)  |
>
> As shown in the table above, PAS demonstrates higher training efficiency compared to AMED, and better sample quality than GITS with negligible additional cost. In contrast, PD, which yields higher sample quality, incurs a significantly larger training cost. Notably, **PAS is theoretically orthogonal to PD, AMED and GITS**, and can further improve their sampling efficiency.
>
>
> >***Theoretical Claims: Lacking theoretical justification for sampling trajectories in a low-dimensional subspace.***
>
> **A**: Inspired by [WV23] and [WV24], we have added more theoretical analysis to the paper (see **Reviewer pbLK, Expand theory**). In brief, since analyzing neural networks directly is inherently intractable, [WV23] and [WV24] approximate diffusion trajectories using Gaussian score structures, providing both theoretical and empirical insights.
>
> Furthermore, [WV23] offers a theoretical analysis in Sec. 4.2, suggesting that diffusion trajectories **resemble 2D rotations on the plane spanned by the initial noise ($x_T$) and the final sample ($x_0$)**. This offers a theoretical justification for why sampling trajectories lie in a low-dimensional subspace. It also explains why trajectories corresponding to different initial noises lie in different subspaces (see our Fig. 2b).
>
>
> >***EDoA1: Add a comparison with other trajectory-based methods (e.g., AMED, GITS) combined with other SOTA solvers (e.g., iPNDM).***
>
> **A**: We did not include additional comparisons between PAS and other trajectory-based methods (e.g., AMED, GITS) primarily because the integration of PAS with DDIM already demonstrates significant improvements in sampling efficiency compared to AMED and GITS, thereby validating the effectiveness of PAS. Furthermore, PAS is theoretically orthogonal to methods like AMED and GITS, and can potentially be combined with them to further enhance sampling efficiency. Therefore, we believe that exploring how to better combine PAS with trajectory-based methods (e.g., AMED, GITS) is a more meaningful direction, which will be considered in our future work.
>
>
> >***Questions For Authors: Have you experimented with alternative dimensionality reduction techniques in place of PCA?***
>
> **A**: We have experimented with both standard PCA (`torch.svd`) and low-rank or sparse matrix PCA (`torch.pca_lowrank`). Notably, PCA_lowrank (used in our paper) computes approximately 1.58× faster than standard PCA, with some loss in accuracy. We provide a comparative experiment on the CIFAR10, as shown in the following table:
>
> PAS+DDIM, FID↓
> | Method\NFE  | 5     | 6     | 8    | 10   |
> | ----------- | ----- | ----- | ---- | ---- |
> | PCA         | 16.38 | 13.08 | 7.01 | 4.39 |
> | PCA_lowrank | 17.13 | 12.11 | 7.07 | 4.37 |
>
> From the table, we can observe that PCA_lowrank and standard PCA yield **comparable results**. Upon further analysis, we believe this is because only around 4 principal components are sufficient to span the complete sampling trajectory space (which consists of 1000 vectors). Since extracting 4 principal components from trajectory space is relatively straightforward, the choice of dimensionality reduction technique is not a critical factor in the effectiveness of the proposed PAS method.
>
>
> [R1] Salimans T et al. Progressive distillation for fast sampling of diffusion models. ICLR 2022.
>
> [R2] Zhou, Zhenyu et al. Fast ode-based sampling for diffusion models in around 5 steps. CVPR 2024.
>
> [R3] Defang Chen et al. On the Trajectory Regularity of ODE-based Diffusion Sampling. ICML 2024.

---

### Official Review · Reviewer_8neu · 2025-03-13

**Overall Recommendation:** 3

**Summary:**

The paper proposes a novel PCA-based Adaptive Search (PAS) method to accelerate diffusion model sampling with minimal additional computational and parameter costs. The key idea of PAS rests on the observation that the sampling trajectory of a parameterized reverse ODE of diffusion model almost lies in a 3D subspace in a high-dimensional space. Thus, PAS involves leveraging Principal Component Analysis (PCA) to identify a small set of orthogonal basis vectors and learns low-dimensional coordinates corresponding to these vectors to correct sampling errors. The main results demonstrate substantial improvements in sampling efficiency and quality with negligible cost. Specifically, PAS reduces the FID score of DDIM from 15.69 to 4.37 on CIFAR10 with 10 sampling steps, using merely around 12 parameters and minimal training time. PAS shows similar enhancements across multiple datasets (e.g., CIFAR10, FFHQ, ImageNet, LSUN Bedroom) and various pre-trained diffusion models, consistently outperforming the state-of-the-art method.

**Claims And Evidence:**

Yes, the paper's main claims are convincingly supported through empirical evaluations and comparative analyses. The paper proposes utilizing PCA to identify a small number of basis vectors spanning the high-dimensional trajectory space of sampling paths. They substantiate this claim with empirical evidence showing that sampling trajectories reside primarily within a low-dimensional subspace. Clear experimental results confirm that correction along these PCA-derived directions is effective in reducing truncation errors.

**Essential References Not Discussed:**

No.

**Experimental Designs Or Analyses:**

Yes, the paper conducts extensive experiments to validate the effectiveness of the proposed method. The authors emphasize minimal computational costs by reporting clearly the training time (e.g., less than 2 minutes on CIFAR10) and the number of parameters (approximately 10), however, direct quantitative performance comparisons against existing training-based methods is lacking. Including such comparisons could better highlight PAS's advantage over a more costly training-based method.

**Methods And Evaluation Criteria:**

Yes, the paper proposes PAS, a lightweight, plug-and-play method to correct discretization errors during diffusion sampling. PAS leverages the inherent geometric property that sampling trajectories lie in a low-dimensional subspace, utilizing PCA to represent and correct sampling directions with minimal parameters efficiently.
The evaluation utilizes standard datasets, such as CIFAR10, FFHQ, ImageNet, LSUN Bedroom, and Stable Diffusion v1.4. The evaluation metric FID objectively measures the quality of generated images. Comparisons against existing state-of-the-art methods (e.g., DPM-Solver++, iPNDM) provide strong evidence for PAS’s effectiveness in terms of improved sampling quality and efficiency at significantly lower parameter and computational costs.

**Other Comments Or Suggestions:**

No.

**Other Strengths And Weaknesses:**

Strengths:

1. The paper introduces a highly original approach by leveraging PCA to significantly reduce the dimensionality of sampling corrections in diffusion probabilistic models, thus effectively addressing computational and storage bottlenecks prevalent in existing methods.
2. The explanations throughout the paper are clear and logically structured.

Weaknesses:

1. While the paper extensively demonstrates empirical effectiveness, it could further clarify theoretical insights behind the PCA-based trajectory correction-particularly why sampling trajectories of all samples exhibit strong consistent geometric characteristics.
2. Discussion of potential limitations regarding the reliance on PCA and robustness to stochastic samplers, remains limited and could be expanded upon.

**Questions For Authors:**

1. The paper highlights the efficiency advantages of PAS over existing training-based methods. Could you provide a more detailed comparison—especially in terms of sampling quality and computational trade-offs—against widely-used training-based methods such as Salimans & Ho (2022)?
2. The PAS demonstrates promising results for deterministic solvers (e.g., DDIM). Could PAS be similarly effective if adapted to stochastic sampling methods? Have you conducted any preliminary experiments or analyses in this direction? Clarification on applicability or limitations would be valuable.

**Relation To Broader Scientific Literature:**

The key contributions of this paper respond to training costs limitations in prior few step generation works, notably the Progressive Distillation method(Salimans & Ho, 2022). While Progressive Distillation substantially improves sampling speed by training distilled models that drastically reduce sampling steps, it faces critical limitations including significant additional training costs and storage demands for the distilled model. This paper addresses these practical bottlenecks by leveraging PCA to identify a compact set of basis vectors spanning the sampling directions, thus requiring minimal retraining and alleviating memory constraints.

**Theoretical Claims:**

This paper do not provide theoretical claims.

---

> ### Author Rebuttal · Authors · 2025-03-30
>
> We sincerely appreciate the reviewer's efforts and valuable review. Below are our responses to all questions. We kindly hope you could consider increasing the score if you are satisfied.
>
> >**Experimental Designs Or Analyses: Direct performance comparisons with training-based methods are lacking, highlighting PAS's advantage.**
>
> **A**: We appreciate your suggestion! We did not include direct performance comparisons because high-cost training-based methods and PAS have their own application scenarios. Training-based methods can achieve one-step sampling but are expensive, making them difficult to apply to large models like Stable Diffusion. In contrast, PAS focuses on high-quality sampling under 10 NFE with negligible cost, making it a **more attractive and practical** solution. Furthermore, we added a comparison of parameter size and training time: Training-based one-step sampling methods require **at least 35.7M** parameters (DDPM, CIFAR10 model), while PAS uses **only ~10**. The table below summarizes training time↓ (A100 hours) on CIFAR10, using the estimation method and certain results from [R1].
>
> **Note**: Training a CIFAR10 model from scratch takes ~200 A100 hours [R1].
> |PD[R2]|Guided PD[R3]|CD[R4]|CTM[R5]|PAS(Ours)|
> |-|-|-|-|-|
> |~195|~146|~1156|~83|<0.04|
>
> >***W1: Theoretical explanation of why the sampling trajectories of all samples exhibit strongly consistent geometric characteristics.***
>
> **A**: Inspired by [WV23] and [WV24], we have added more theoretical analysis to the paper (**Reviewer pbLK, Expand theory**). Briefly, since analyzing neural networks directly is inherently intractable, [WV23] and [WV24] approximate diffusion trajectories using Gaussian score structures, providing both theoretical and empirical insights.
>
> Specifically, [WV24] derives an analytical form of the EDM trajectory in Eq. 15(Gaussian structure):
>
> $x_t=\mu+\frac{\sigma_t}{\sigma_T} x_T^\perp +\sum_{k=1}^r \psi (t,\lambda_k)c_k(T)u_k$,
>
> which shows that $x_t$ is a linear combination of $x_T^\perp$ and $\mu, u_k$. Here, $\mu, u_k$ are the dataset-dependent mean and basis vectors that define the data manifold, while $x_T^\perp$ is an off-manifold component. Furthermore, given $x_T$, $c_k(T)$ is a constant, and the rate of change of the coefficients for the vectors $x_T^\perp, \mu$, and $u_k$ depends only on the dataset and the timestep $t$. For a fixed dataset, as $\sigma_t$ decreases during the sampling process, each sample is constrained by the same temporal decay scale, converging to the data manifold. This theory heuristically explains why all samples exhibit strongly consistent geometric characteristics.
>
> >***W2 and Q2: Discuss the limitations of the reliance on PCA and the robustness of PAS to stochastic samplers.***
>
> **A**: The effectiveness of PAS relies on the accuracy of the basis derived from PCA, which may be influenced by noise from stochastic samplers. To further explore this, we added experiments using PAS on the stochastic sampler of EDM [R6] without the 2-order correction ([R6] Algorithm 2) on CIFAR10, with the following results:
>
> FID↓
> |Method\NFE|5|8|10|20|
> |-|-|-|-|-|
> |Stochasticity|55.71|27.57|21.48|10.95|
> |+PAS|41.44|23.81|19.04|10.48|
>
> The table clearly shows that PAS is **equally effective** for the stochastic sampler, further highlighting the robustness of PAS. Nonetheless, in general, SDE has a higher quality upper bound, with 1000 NFE outperforming ODE. But for sampling with fewer steps, ODE performs better. Therefore, PAS with ODE would result in faster sampling speeds.
>
> >***Q1: A comparison of sampling quality and computational trade-offs with training-based methods like PD[R2].***
>
> **A**: We have added some comparative results on the training cost and sample quality between PD and PAS, as shown in the following tables:
>
> FID↓(A100 hours↓)
> |Method\NFE|1|2|4|8|
> |-|-|-|-|-|
> |DDIM+PD[R2]|9.12(~195)|4.51(~171)|3.00(~159)|2.57(~146)|
>
> |Method\NFE|5|6|8|10|
> |-|-|-|-|-|
> |DDIM+PAS|17.13(~0.01)|12.11(~0.02)|7.07(~0.03)|4.37(~0.04)|
> |iPNDM+PAS|13.61(~0.01)|7.47(~0.02)|3.87(~0.03)|2.84(~0.04)|
>
> As shown in the tables above, PD requires ~146 A100 hours of training time for 8 NFE to achieve 2.57 FID, whereas PAS with iPNDM requires **only 0.04 A100 hours with 8 parameters** to achieve 2.84 FID at 10 NFE. Therefore, PD and PAS each have their own application scenarios. PAS achieves impressive performance with negligible cost, making it a **more attractive and practical** solution.
>
> [R1] Zhou Z et al. Simple and fast distillation of diffusion models. NeurIPS 2024.
>
> [R2] Salimans T et al. Progressive distillation for fast sampling of diffusion models. ICLR 2022.
>
> [R3] Meng C et al. On distillation of guided diffusion models. CVPR 2023.
>
> [R4] Song Y et al. Consistency models. ICML 2023.
>
> [R5] Kim D et al. Consistency trajectory models: Learning probability flow ode trajectory of diffusion. arXiv:2310.02279.
>
> [R6] Tero Karras et al. Elucidating the design space of diffusion-based generative models. NeurIPS 2022.

---

> > ### Comment · Reviewer_8neu · 2025-04-09
> >
> > Thanks the authors for the rebuttal. My concerns are addressed. I recommend acceptance as all reviewers agreed.

---

> > > ### Author Response · Authors · 2025-04-09
> > >
> > > We are so glad to hear that your concerns have been addressed! Thank you once again for recognizing our work!

---

### Official Review · Reviewer_pbLK · 2025-03-13

**Overall Recommendation:** 3

**Summary:**

The authors leveraged the previous finding that the diffusion sampling trajectories are low dimensional, and that part of it is more curvy. Then they developed a method to learn the PCA basis of current ongoing trajectory during sampling and then learn coefficient to recombine the PC vectors to correct for the truncation errors in the small NFE solvers. They can train the PC correction coefficients efficiently on a bunch of “ground truth” trajectory with higher NFEs, and obtain corrections for sampler with fewer NFEs, effectively distilled out a dataset specific fast sampler based on the PC correction coefficients.

The method is evaluated on CIFAR10 32x32 to Stable diffusion 512, and showed effectiveness through the scales.

## Update after rebuttal
The authors showed a close comparison with the Gaussian approximation based teleportation / acceleration method in [WV24], and an updated benchmark. The updated theoretical link between the low dimensional trajectory to the Gaussian theory also made it stronger.
The reviewer is happy to maintain the score as such.

**Claims And Evidence:**

Most claims are clear and supported.

Some interpretation of why the method works could be made clearer.

- **Minor point** “*Thus, we can infer that the sampling trajectory first appears linear, then transitions to a curve, and ultimately becomes linear again under the attraction of a certain mode.*”
In Fig. 3a the authors interpreted the S shape truncation error as the error at middle range is due to higher curvature. This is possible but my interpretation is the earlier trajectory is quite linear, and later trajectory may not be linear but took smaller steps, so the L2 error do not increase that much (See [KMTS22] Fig.3 [WV23] Fig.1). If the author do want to consolidate the curvature intuition, you could directly perform an empirical analysis of how much the trajectory rotate at each part.
- The failure of ablation study without Adaptive search is worth noticing, from the FID, it seems it failed spectacularly.
I’m curious about the author’s interpretation of it (e.g. curvature). My interpretation is that the PCA basis you got from some part of the trajectory (e.g. see Fig. 18 in [WV23]) may be not good, or too contaminated with noise, so not useful to correct the trajectory. Thus those PC may not generally help correct for truncation error.
I think a deeper analysis of that part may illustrate why this method may or may not work.

[KMTS22] Karras, T., Aittala, M., Aila, T., & Laine, S. (2022). Elucidating the design space of diffusion-based generative models. Advances in neural information processing systems, 35, 26565-26577.

[WV23] Wang, B., & Vastola, J. J. (2023). Diffusion models generate images like painters: an analytical theory of outline first, details later.

**Essential References Not Discussed:**

**Methods**

- Regarding Figure 2 results of PCA of sampling trajectory, one reference the authors missed is [WV23], where they also systematically studied the PC structure of sampling trajectory and provided a theoretical account explaining the low dimensional (mostly 2d) nature of the sampling trajectory. Basically they showed the low dim subspace is the one spanned by the initial noise and the final image sample, which explains your Fig.2B, since different trajectory starts from different initial noise, they won’t have a shared subspace / low dimensional structure.

    Authors of [WV23] also found a better way to understand these trajectories is to look at the PCs of the projected outcome x0_hat, their PC should be closer to those of the image manifold. Seems Fig 2. in [WV23] and Fig.1 in your paper has interesting connections.

    The authors should mention these results in line 156-160 “*This indicates that the entire sampling trajectory lies in a three-dimensional subspace embedded …* ”

- The theory and observations of early diffusion trajectory is well approximated by Gaussian score trajectory leads to some strong predictions about what the PC’s you are getting, they should somehow be aligned with the PC of target data manifold [WV23]. So the authors could discuss those as further theoretical justifications of the method.

**Results**

- In the main benchmark table 2, it seems the results the authors get from PAS-acceleration are similar or sometimes slightly worse than the results from a recent paper [WV24] Fig.15, which also achieves accelerated sampling via analytical teleportation, based on the PCA of data manifold without any retraining. They also tested on EDM pretrained on CIFAR and FFHQ, AFHQ, so the NFE and FID numbers are totally comparable to those in your table. The authors could consider adding them to the benchmark.
    - I have some suspicion that the mechanism of acceleration may be shared, i.e. due to the Gaussian score structure in diffusion. Due to the Gaussian structure, the trajectory from any initial noise state can be analytically predicted based on mean and PC of data, so they can teleport the solution. I guess the mechanism of your PAS acceleration is likely to be related to that. The authors might benefit from discussing the connections.
    - On the other hand, one difference is, their acceleration mainly happen at earlier part of trajectory (e.g. high noise regime) and from Table 1, seems PAS works by correcting more at the low noise regime, suggesting different correcting mechanism.
    - Mentioning this is not to reduce the contribution or novelty of the current result. I think their methods were hard to be applied to Stable Diffusion or very high res images, due to they need to compute PCA of the dataset. Trajectory based PCA and training seems to circumvent a bit.

[WV23] Wang, B., & Vastola, J. J. (2023). Diffusion models generate images like painters: an analytical theory of outline first, details later.

[WV24] Wang, B., & Vastola, J. J. (2024). The Unreasonable Effectiveness of Gaussian Score Approximation for Diffusion Models and its Applications. arXiv preprint arXiv:2412.09726.

**Experimental Designs Or Analyses:**

Standard design and analysis.

**Methods And Evaluation Criteria:**

The experimental set up and evaluations are quite standard and performed well!

- There are some very recent results the authors didn’t include in the benchmark, which could have some shared mechanism with the paper. [WV24] (detailed in missing reference part.)
- One aspects the authors didn’t mention is that even though training is super efficient, obtaining the sampling trajectory for training is likely not efficient. In your case obtaining 5k to 10k sampling trajectory with high NFE could take some time. The authors could mention the time of obtaining such trajectories in additional to training (around L311-318).
This is actually a bit similar to [WV24], where their acceleration is parameter / training free, but to sample the data to compute PCA takes time, if they don’t have access to training set.

[WV24] Wang, B., & Vastola, J. J. (2024). The Unreasonable Effectiveness of Gaussian Score Approximation for Diffusion Models and its Applications. arXiv preprint arXiv:2412.09726.

**Other Comments Or Suggestions:**

- Table 1 caption is not very clear, authors could more clearly annotate what are the list of numbers is showed in the table.

**Other Strengths And Weaknesses:**

- The method is simple to implement and parameter and compute efficient, also conceptually beautiful.
- The illustrations and tables were well made and authors put efforts into explaining their ideas.

**Questions For Authors:**

N.A.

**Relation To Broader Scientific Literature:**

See below.

**Theoretical Claims:**

There is not much theoretical claims in this paper. The intuition of correcting the more curvy part of trajectory is interesting, the authors could provide a slightly more formal treatment of the

Though more connection to some theoretical framework can better frame or motivate the method. [WV24]

---

> ### Author Rebuttal · Authors · 2025-03-30
>
> We sincerely appreciate the reviewer's detailed review and insightful suggestions!
>
> **Abbreviation: CaE (Claims And Evidence), MaEC (Methods And Evaluation Criteria), ERND (Essential References Not Discussed)**
>
> >***CaE1: The later trajectory may not be linear but instead takes smaller steps.***
>
> **A**: Your interpretation is reasonable. However, [KMTS22] and [WV23] only analyze the trajectory in 2D, which may lose some information. [R1] visualizes trajectory in 3D ([R1] Fig. 4) and offers a "boomerang" explanation (linear-nonlinear-linear) in Sec. 3, showing that the later part is indeed linear. We have added a detailed analysis in the revised version.
>
> [R1] Chen D et al. On the trajectory regularity of ode-based diffusion sampling. ICML2024.
>
> >***CaE2: The degradation without adaptive search may be due to the obtained PCs not being good.***
>
> **A**: Notably, this paper has conducted a similar analysis in lines 377-384(Right): "The errors in the linear part are negligible; the correction does not further reduce errors and instead introduces noise", highlighting the **necessity and effectiveness of the adaptive search**.
>
> >***Expand theory (MaEC1, Theoretical Claims, and ERND_M1): The study misses citing [WV23] and [WV24], which could theoretically motivate Fig. 2b and the correction of the more curved parts(Fig. 1).***
>
> **A**: We have carefully revisited [WV23] and [WV24], which approximate diffusion trajectories by Gaussian score structure and provide both theoretical and empirical insights. These works are indeed inspiring and insightful!
>
> Specifically, [WV23] offers a theoretical analysis in Sec. 4.2, suggesting that diffusion trajectories resemble 2D rotations on the plane spanned by the initial noise ($x_T$) and the final sample ($x_0$). This provides a theoretical underpinning for our Fig. 2b, where trajectories corresponding to different initial noises lie in different subspaces.
>
> **Correction of the more curved parts**: [WV24] derives an analytical form of the EDM trajectory in Eq. 15 (Gaussian structure):
>
> $x_t=\mu+\frac{\sigma_t}{\sigma_T} x_T^\perp +\sum_{k=1}^r \psi (t,\lambda_k)c_k(T)u_k$,
>
> which shows that $x_t$ is a linear combination of $x_T^\perp$ and $\mu,u_k$. PAS is designed to learn statistical patterns of the dataset that are independent of the initial noise $x_T$, i.e., the coefficients of $\mu,u_k$. As $\sigma_t \to 0$, the influence of $x_T^\perp$ on $x_t$ diminishes. So, in the low-noise regime (i.e., more curved parts), PAS batter captures the PCs of the data ($\mu,u_k$), leading to more accurate correction. This offers a theoretical explanation for why PAS tends to work better in the more curved regions (Fig. 1). We have added relevant discussions in Sec. 3 and the Related Work of the revised manuscript. The alignment between PAS and the conclusions of [WV23], [WV24] is indeed intriguing!
>
> >***MaEC2: Discuss the time taken to obtain the sampling trajectory for training (similar to [WV24]).***
>
> A: We did not discuss the cost because other training-based methods also require sampling full or partial trajectory for training. In the revised version, we added a discussion on the time taken: 3.26m for CIFAR10 and 1.79h for Bedroom256 with 50 NFE and a 5k trajectory on an A100 GPU.
>
> >***ERND_M2 and ERND_R1-3: Discuss the connections and differences between PAS and [WV23], [WV24], and include them as a benchmark.***
>
> **A**: The proposed PAS corrects sampling in the low-noise regime, while [WV24] accelerates sampling in the high-noise regime. So, combining [WV24] with PAS is indeed interesting. Theoretically, we can analytically warm up the PCs from the Gaussian structure and then **correct the sampling starting from the teleported solution** $x_{t'}$, further improving the FID in [WV24] Fig. 15. However, [WV24] is limited by the computational cost of estimating means and covariances on large pixel datasets. Furthermore, another key challenge is how to effectively combine the Gaussian (analytical but imprecise under low noise) and neural (precise but costly) structure to obtain more exact PCs with minimal cost. This will be a focus of our future work.
>
> >***ERND_R4: The PAS is hard to apply to very high-res images.***
>
> **A:** No, the PAS can **be easily applied to high-res images**. This is because we use `torch.pca_lowrank` to decompose the trajectory matrix as $\mathbf{X}^{\prime}\in \mathbb{R}^{(N-i+2)\times D}$ (Eq. 13), where $i\leq N,N\leq 10$, so its dimension is low and the rank $r\ll D$. For Stable Diffusion(SD), $D$=64x64x4 from the latent space. In the table below, we compare the time for 1PCA and 1NFE, and the PCA time is negligible.
>
> pre 128 samples time↓ (s)
> | |SD|Bedrooom256|
> |-|-|-|
> |1NFE|30.20|10.05|
> |1PCA|0.06|0.2|
>
> >***Other Suggestions: Table1 caption needs clarification.***
>
> **A**: Thanks for pointing out the issues; we have revised them. Briefly, the list of numbers denotes all the time points requiring correction from adaptive search, with each element $i\in [N,1]$.

---

> > ### Comment · Reviewer_pbLK · 2025-04-07
> >
> > The authors’ willingness to revisit [WV23] and [WV24] and clarify the connection between the current method and established theoretical results is much appreciated. This connection greatly benefits readers by providing deeper insight into the methodology. Thanks for mentioning the nice paper [R1] which also complements the view from [WV23].
> >
> > I am particularly enthusiastic about the proposed future direction of using analytical teleportation as a warm-up phase, followed by PAS to correct the more curved segments of the trajectory.
> > However, it might still be helpful to **include the teleportation method in the benchmark tables**, as its contribution to achieving low FID scores would offer readers a more comprehensive view. I agree that combining these techniques will lead to improved FID outcomes.
> >
> > Additionally, I concur that PAC appears more scalable to high-resolution images on complex datasets. This scalability stems from its approach of factorizing only the current trajectory, in contrast to analytical teleportation, which requires pca_lowrank on a larger set of training images.
> >
> > Overall, the paper is clean and highly informative—kudos to the authors for their excellent work!

---

> > > ### Author Response · Authors · 2025-04-08
> > >
> > > Thank you very much for your response and the recognition of our work! We are so glad to hear that the rebuttal helped clarify how the proposed PAS can be easily applied to high-resolution images on complex datasets. We also agree that combining analytical teleportation with PAS and including the results in the benchmark tables is beneficial. Per your suggestion, we have added additional experiments on the CIFAR10, as shown below:
> > >
> > > We use $\sigma\_{skip} = 10.0$ for teleportation in equation ([WV24] Algorithm 1):
> > >
> > > $\mathbf{x}_{t^{\prime}} \leftarrow \boldsymbol{\mu}+\frac{\sigma\_{skip}}{\sigma\_{\max }}\left(\mathbf{I}-\mathbf{U} \mathbf{U}^{T}\right)\left(\mathbf{x}\_{T}-\boldsymbol{\mu}\right)+\sum\_{k=1}^{r} \sqrt{\frac{\sigma\_{s k i p}^{2}+\lambda\_{k}}{\sigma\_{\max }^{2}+\lambda\_{k}}} \mathbf{u}\_{k} \mathbf{u}\_{k}^{T}\left(\mathbf{x}\_{T}-\boldsymbol{\mu}\right)$.
> > >
> > > CIFAR10, FID↓
> > >
> > > | Method\NFE                   | 5        | 6        | 8        | 10       |
> > > | ---------------------------- | -------- | -------- | -------- | -------- |
> > > | DDIM                         | 49.68    | 35.63    | 22.32    | 15.69    |
> > > | + teleportation              | 24.50    | 18.41    | 12.04    | 8.78     |
> > > | + PAS (**Ours**)                 | 17.13    | 12.11    | 7.07     | 4.37     |
> > > | + teleportation + PAS (**Ours**) | **9.15** | **5.16** | **3.65** | **3.16** |
> > > |                              |          |          |          |          |
> > > | iPNDM                        | 16.55    | 9.74     | 5.23     | 3.69     |
> > > | + teleportation              | 7.25     | 4.89     | 3.08     | 2.49     |
> > > | + PAS (**Ours**)                 | 13.61    | 7.47     | 3.87     | 2.84     |
> > > | + teleportation + PAS (**Ours**) | **5.16** | **3.76** | **2.77** | **2.40** |
> > >
> > > As shown in the table, PAS can **indeed** be combined with teleportation to **further improve FID** outcomes. We observe that **iPNDM + teleportation + PAS** outperforms the results reported in [WV24] Fig. 15 for NFE $\in$ {5, 6, 8, 10} on CIFAR10. These results have been included in the benchmark tables (our Tab. 2). Your comments also motivate us to further explore the potential of hyperparameter settings (e.g., $\sigma_{skip}$) in the combination of PAS and teleportation, as well as evaluate their combination with more datasets and solvers (e.g., DPM-Solver-v3 [R2]). More results will be incorporated into the benchmark tables in the revised version of the paper.
> > >
> > > Thank you again for your meticulous review and insightful comments, which have greatly improved our paper!
> > >
> > > [R2] Zheng K et al. Dpm-solver-v3: Improved diffusion ode solver with empirical model statistics. NeurIPS 2023.

---

### Decision · Program_Chairs · 2025-05-01

**Decision:**

Accept (poster)

**Comment:**

The paper introduces a training paradigm called PAS (PCA-based Adaptive Search), for accelerating DPMs (Diffusion Probabilistic Models) with minimal training costs and learnable parameters. In particular, the authors propose to obtain a few basis vectors via PCA, and then learn their low-dimensional coordinates to correct the high-dimensional sampling direction vectors. Numerical experiments are provided to illustrate the method.

Reviewers generally agree that this is an original approach and appreciate the provided numerical experiments, which clearly demonstrate the effectiveness of the proposed method. All reviewers point out the lack of theoretical analysis in the current manuscript. In their response, the authors pointed out connection with the Gaussian  approximation based teleportation / acceleration method in Wang and Vastola (2024), which
reviewers pbLK and 8neu found to be satisfactory.

While the paper is lacking theory, I believe the proposed method would be of interest and agree with the acceptance recommendation by all reviewers.

Note: The authors claim that the code will be available. This should be released upon acceptance of the paper.